# Axonemal structures reveal mechanoregulatory and disease mechanisms

Travis Walton[1,11], Miao Gui[1,10,11], Simona Velkova[2], Mahmoud R. Fassad[2,3], Robert A. Hirst[4], Eric Haarman[5], Christopher O'Callaghan[6], Mathieu Bottier[7,8], Thomas Burgoyne[7,9], Hannah M. Mitchison[2] & Alan Brown[1✉]

Motile cilia and flagella beat rhythmically on the surface of cells to power the flow of fluid and to enable spermatozoa and unicellular eukaryotes to swim. In humans, defective ciliary motility can lead to male infertility and a congenital disorder called primary ciliary dyskinesia (PCD), in which impaired clearance of mucus by the cilia causes chronic respiratory infections[1]. Ciliary movement is generated by the axoneme, a molecular machine consisting of microtubules, ATP-powered dynein motors and regulatory complexes[2]. The size and complexity of the axoneme has so far prevented the development of an atomic model, hindering efforts to understand how it functions. Here we capitalize on recent developments in artificial intelligence-enabled structure prediction and cryo-electron microscopy (cryo-EM) to determine the structure of the 96-nm modular repeats of axonemes from the flagella of the alga *Chlamydomonas reinhardtii* and human respiratory cilia. Our atomic models provide insights into the conservation and specialization of axonemes, the interconnectivity between dyneins and their regulators, and the mechanisms that maintain axonemal periodicity. Correlated conformational changes in mechanoregulatory complexes with their associated axonemal dynein motors provide a mechanism for the long-hypothesized mechanotransduction pathway to regulate ciliary motility. Structures of respiratory-cilia doublet microtubules from four individuals with PCD reveal how the loss of individual docking factors can selectively eradicate periodically repeating structures.

The prototypic axoneme of a motile cilium consists of a cylindrical arrangement of doublet microtubules (DMTs) surrounding a central apparatus comprising two singlet microtubules. DMTs are organized into 96-nm repeating units, which cryo-electron tomography (cryo-ET) studies have shown to be a landscape of interconnected dynein motors and regulatory complexes[2]. In total, there are 11 dynein motors in a single 96-nm repeat unit, and they are classified according to their location (inner and outer) and the number of heavy chains (HCs) they have. Each HC contains an ATPase 'head' domain and a helical 'tail' domain that associates with intermediate-chain (IC) and light-chain (LC) subunits. In a 96-nm repeat, there are six different single-headed inner dynein arms (IDA*a*–IDA*e* and IDA*g*), one double-headed inner dynein arm (IDA*f*) and four identical multi-headed outer dynein arms (ODAs). These ODAs, numbered 1–4 in the proximal to distal direction[3], are organized with 24-nm periodicity in a linear array with extensive head-to-tail interactions[4–6]. During ciliary beating, axonemal dyneins docked onto one DMT interact and pull against a second, neighbouring DMT through their motor domains. ODAs provide most of the pulling

power, and the six single-headed IDAs are thought to provide torque between DMT pairs[7–9]. Axonemal dynein activity switches between specific DMTs on opposite sides of the axoneme[10] through a hypothesized mechanoregulatory pathway involving the central apparatus, DMT-bound T-shaped radial spokes (RSs), the nexin–dynein regulatory complex (N-DRC) and IDA*f*[11].

Despite extensive genetic and tomographic data, the detailed structures of axonemes, and hence their interactions and mechanoregulatory mechanisms, remain poorly understood. We used single-particle cryo-electron microscopy (cryo-EM) and integrative modelling to build atomic models of the 96-nm modular repeats of axonemes from *Chlamydomonas reinhardtii* flagella and human respiratory cilia.

## The *C. reinhardtii* axoneme

To determine the structure of the modular repeat of the *C. reinhardtii* axoneme, we reconstructed cryo-EM maps of DMTs from splayed axonemes (Methods and Supplementary Fig. 1). Computational limits

[1]Department of Biological Chemistry and Molecular Pharmacology, Harvard Medical School, Boston, MA, USA. [2]Genetics and Genomic Medicine Department, UCL Great Ormond Street Institute of Child Health, University College London, London, UK. [3]Department of Human Genetics, Medical Research Institute, Alexandria University, Alexandria, Egypt. [4]Centre for PCD Diagnosis and Research, Department of Respiratory Sciences, University of Leicester, Leicester, UK. [5]Department of Pediatric Respiratory Medicine and Allergy, Emma Children's Hospital, Amsterdam University Medical Centers, Amsterdam, The Netherlands. [6]Infection, Immunity & Inflammation Department, NIHR GOSH BRC, UCL Great Ormond Street Institute of Child Health, University College London, London, UK. [7]Royal Brompton Hospital, Guy's and St Thomas' NHS Foundation Trust, London, UK. [8]National Heart and Lung Institute, Imperial College London, London, UK. [9]Institute of Ophthalmology, University College London, London, UK. [10]Present address: Liangzhu Laboratory, Zhejiang University, Hangzhou, China. [11]These authors contributed equally: Travis Walton, Miao Gui. ✉e-mail: alan_brown@hms.harvard.edu

imposed by the large size of the repeat meant that we focused on individual axonemal complexes by particle subtraction, refinement and classification before reconstituting the full 96-nm repeat in silico. For some regions, this approach led to structures with resolutions better than 4 Å, whereas other regions had lower resolution because of flexibility that impaired the ability to successfully align the particles, loss of molecular complexes during sample preparation and averaging of asymmetric features (Supplementary Figs. 1–9). The resolution of these maps exceeds what has been achieved previously (about 25 Å) by cryo-ET of *C. reinhardtii* axonemes[12].

To interpret the maps and create atomic models, we used an integrative approach that combined de novo modelling, artificial intelligence (AI)-based structure prediction (Supplementary Figs. 10–12), cryo-ET studies of *C. reinhardtii* mutants and previous cryo-EM-derived models of doublet microtubule inner proteins (MIPs)[13], RSs[14] and the ODA[5] (Methods). Together with a structure of the central apparatus[15], we generate the most complete atomic model of an axoneme thus far (Fig. 1 and Supplementary Video 1). For all complexes, the subunits identified are provided in Supplementary Table 1 and the rationale for their placement is in Supplementary Table 2. Model statistics are provided in Extended Data Table 1.

## Structures of axonemal dyneins

The model includes all three classes of axonemal dynein: the triple-headed ODA, the regulatory double-headed IDAf and all six single-headed IDAs (IDAa–IDAe and IDAg). Although we previously reported[5] a structure of the ODA, the improved accuracy of AI-enabled structure prediction and an atomic model[6] of the ODA from the unicellular ciliate *Tetrahymena thermophila* allowed us to identify two thioredoxin-like subunits (LC3 and LC5) and a second, algal-specific Kelch-like β-propeller domain in the α-HC tail (Extended Data Fig. 1a). We also built an atomic model of IDAf (Fig. 2a), which has two ATPase motor domains (fα and fβ) but displays no or little motility in vitro[16–18]. Despite their differences in activity, our structure demonstrates that the IDAf core (defined as the tails of the HCs and the ICs and LCs with which they associate) closely resembles the ODA core[5] (Extended Data Fig. 1b). Both have an IC–LC block containing two β-propeller-containing ICs, a heterodimer of LC7a–LC7b, three LC8-like dimers and a heterodimer of the TCTEX family. Two IDAf-specific subunits, IC97 (ref. 19) and FAP120 (ref. 20), expand this region compared with the ODA (Extended Data Fig. 1b). Relative to the core, the motor domains of IDAf are rotated and displaced compared with those of the ODA (Extended Data Fig. 1c), probably allowing IDAf to bind the neighbouring DMT from its unique position in the 96-nm repeat.

## Multipartite docking of IDAf

IDAf is attached to the DMT through a multipartite docking mechanism involving four different anchors (Fig. 2a,b). The first anchor, which links to ODA1, will be described in detail later (Outer-inner dynein links section). The second anchor corresponds to Modifier of inner arms (MIA), a heterodimeric coiled coil of FAP73 and FAP100 (FAP73/100; ref. 21) with two separate attachments to the DMT surface (Extended Data Fig. 2a,b). MIA bisects the HC tails of IDAf and therefore resembles the DC1–DC2 (DC1/2) coiled coil of the ODA docking complex (ODA-DC), which bisects the γ- and β-HCs of the ODA[5] (Extended Data Fig. 2c). Because IDAf has multiple attachment points, MIA is not essential for docking IDAf to DMTs[21], but DC1/2 is required for the docking of ODAs[22]. The third anchor of IDAf corresponds to a dimer of FAP57 or a member of the FAP57 family[23]. The FAP57 dimer forms a stack of four β-propellers that contacts the fα tail (Fig. 2a and Extended Data Fig. 2d), consistent with the proposed location of FAP57 based on subtomogram averaging[23]. The fourth and final anchor is the

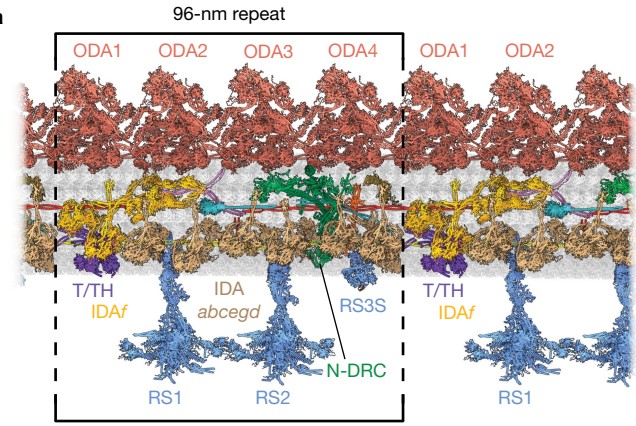

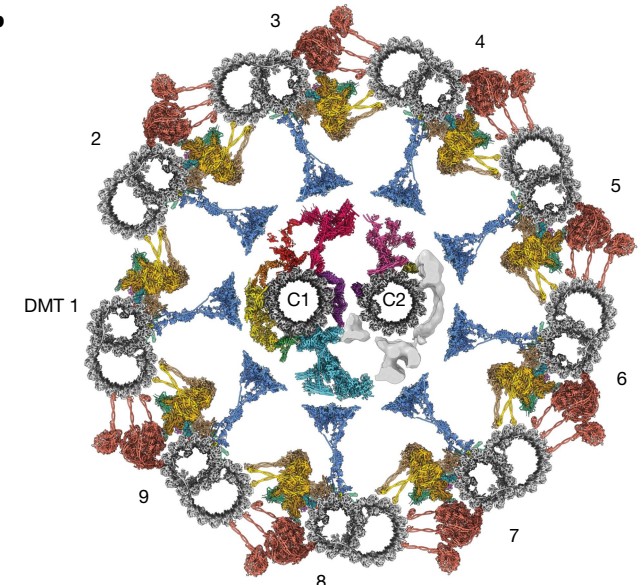

**Fig. 1 | Atomic model of the modular repeat of the *C. reinhardtii* axoneme.** **a**, A longitudinal section of an atomic model of a fully decorated DMT showing the 96-nm repeat boxed. **b**, Cross-sectional model of the *C. reinhardtii* axoneme composed of nine copies of the 96-nm repeat revealed by our study and the C1 and C2 microtubules of the central apparatus (PDB: 7SQC and 7SOM)[15]. The model combines data from multiple sources and provides a consensus view that does not fully capture the asymmetries observed in vivo.

tether–tetherhead (T/TH) complex[24,25], a stellate structure formed by the entwining of two structurally similar proteins, FAP43 and FAP44 (Extended Data Fig. 3a,b). Dimerization of FAP43 and FAP44 explains why genetic ablation of either factor leads to the loss of the other from mutant cilia[26]. MOT7, a protein reported to regulate *C. reinhardtii* IDAf activity in response to blue light[12], occupies the centre of the stellate structure (Extended Data Fig. 3c,d). The N terminus of FAP44 binds the fα motor through a pair of β-propellers (Extended Data Fig. 3a) and its C terminus forms a coiled coil with FAP43 across the DMT surface, terminating at the inner junction between the A and B tubules (Extended Data Fig. 3e). The T/TH-mediated connection between IDAf and the DMT probably constrains IDAf mobility during the powerstroke cycle[24] and undergoes a conformational change in actively beating cilia[10].

## N-DRC forms electrostatic DMT links

Our atomic model shows that the 11-subunit N-DRC is built around long, bifurcated coiled coils that are joined together at a 'bulb' on the

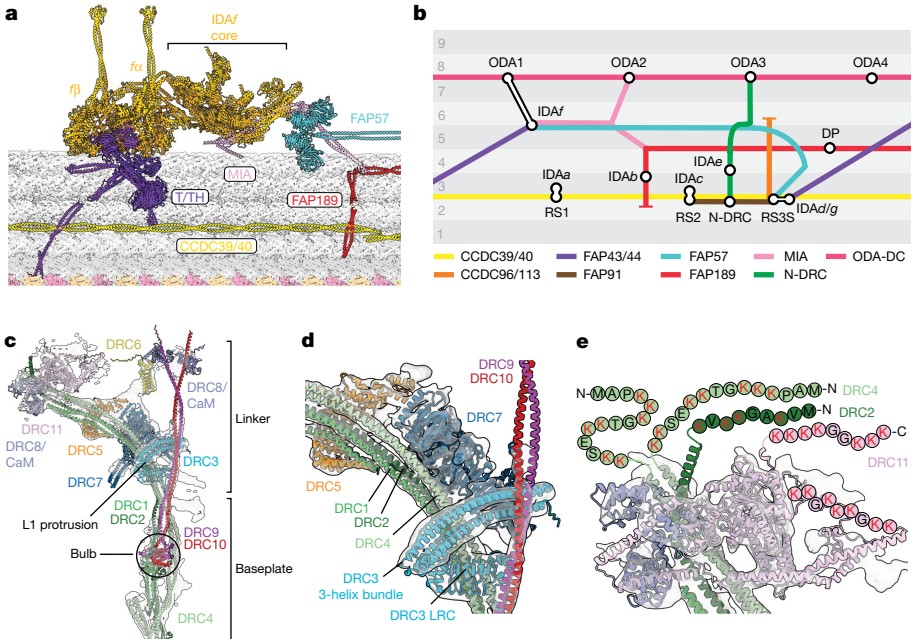

**Fig. 2 | Structures of mechanoregulatory complexes from the *C. reinhardtii* axoneme. a**, Atomic model depicting the multipartite tethering of IDA*f* by the MIA complex (FAP73/100), a FAP57 homodimer and T/TH. Coiled coils of CCDC39/40 and FAP189 interact with T/TH and MIA, respectively. **b**, A conceptual schematic, in the style of a London Underground map, showing the interconnectivity between axonemal complexes in a 96-nm repeat. Circles represent axonemal complexes, coloured lines represent the path of docking factors and alternating grey zones represent protofilaments. DP, distal protrusion. **c**, Composite cryo-EM map and model of the N-DRC. CaM, calmodulin. **d**, The position of DRC3 at a fork in the N-DRC structure. **e**, DRC2, DRC4 and DRC11 position flexible, lysine-rich regions towards the neighbouring DMT. Individual letters indicate amino acids.

microtubule-bound baseplate and fork at the N-terminal leucine-rich repeat (LRR) domain of DRC3 (Fig. 2c,d and Extended Data Fig. 4a,b). After the fork, coiled coils of DRC1–DRC2 and DRC4–DRC4 curve proximally, providing sequential binding sites for DRC7, DRC5 and DRC11. These positions agree with subtomogram averages of *C. reinhardtii* mutant axonemes[27] lacking DRC7 and DRC11 (Extended Data Fig. 4c–e) and structural-labelling approaches[28] that positioned DRC1, DRC2, DRC4 and DRC5. The elongated architecture orients six N-DRC subunits with lysine-rich regions in either flexible loops or termini at the interface with the neighbouring DMT. Three of these subunits, DRC2, DRC4 and DRC11, form a cluster on the proximal tip of the N-DRC that contains at least 60 lysine residues (Fig. 2e). On the distal tip, the termini of DRC6, DRC9 and DRC10 point an additional 26 lysine residues towards the neighbouring DMT. These positively charged side chains probably interact electrostatically with negatively charged polyglutamylate post-translational modifications of the neighbouring B-tubule protofilaments[29] to form a physical link with its neighbouring DMT[30]. Molecules of calmodulin (CaM) and/or the calmodulin-like DRC8 subunit bound to IQ motifs on DRC9 to DRC11 near the interface (Fig. 2c,e) suggest that calcium may also regulate the interaction between neighbouring DMTs.

## Outer–inner dynein links (OIDLs)

As well as making contact with their neighbouring DMT, both IDA*f* and the N-DRC interact with ODAs and IDAs, forming multiple OIDLs that have a proposed regulatory function to coordinate axonemal dynein activity across the 96-nm repeat unit. IDA*f* contacts both ODA1 and ODA2, and N-DRC contacts ODA3 (ref. 3; Fig. 3a). The OIDL in each case is mediated by the TCTEX heterodimer (LC2–LC7) of the ODA core, implicating this region as a critical regulatory contact point. ODA1 connects to IDA*f* through the β-propeller domain of IC140 and nearby helical bundles of *f*β (Fig. 3b). This connection has been observed in all axonemes studied thus far, including those with double-headed

ODAs[10,31,32]. The second OIDL involves a helical segment of FAP100 that branches off MIA (Fig. 3c). Without MIA, *C. reinhardtii* mutants swim slowly with a low flagellar beat frequency[21,33]. Whether lower motility is due to reduced IDA*f* anchoring or the loss of OIDL2 remains to be investigated. OIDL3 involves the LRR domain of the N-DRC subunit DRC7 (Fig. 3d). Mammalian homologues of DRC7 lack the LRR domain (Fig. 3d), suggesting that this OIDL is specific to algae. We did not observe a connection between ODA4 and the adjacent distal protrusion[3], a 155-Å-tall armadillo repeat protein that we identify as FAP78 (Extended Data Fig. 5a–d).

We also observed that the N-DRC specifically contacts IDA*g* through DRC3, consistent with earlier subtomogram averages of *drc3* mutant axonemes[34]. Specifically, a three-helix bundle of DRC3 contacts the linker of the IDA*g* motor (Fig. 3e). The linker of dynein motors bends during the powerstroke cycle, suggesting that the interaction between DRC3 and IDA*g* may regulate dynein activity. Alternatively, this interaction may also help position or rotate the IDA*g* motor to alter its binding to the neighbouring DMT, as proposed for the regulation of IDA*c* by RS2 (ref. 14). For IDA*f*, we did not observe a direct interaction with IDA*a*, although the TCTEX heterodimer of IDA*f* comes close to the linker of its motor domain, suggesting that they could interact in a beating axoneme.

## OIDLs break inter-ODA contacts

During our analysis of OIDLs, we observed multiple conformations of IDA*f* and N-DRC, consistent with their proposed roles as mechanoregulators of axonemal dynein activity. The core of IDA*f* rotates 6° relative to the DMT (Supplementary Video 2), whereas the N-DRC linker tilts by 9° (Supplementary Video 3). Preservation of the OIDLs during the movement of IDA*f* and N-DRC draws the IC–LC block and γ-HC motor of their partner ODAs forwards by similar amounts (about 20 Å) (Fig. 3f). This movement results in the breaking of inter-ODA contacts between the γ-HC motor and the tail domains of its proximal

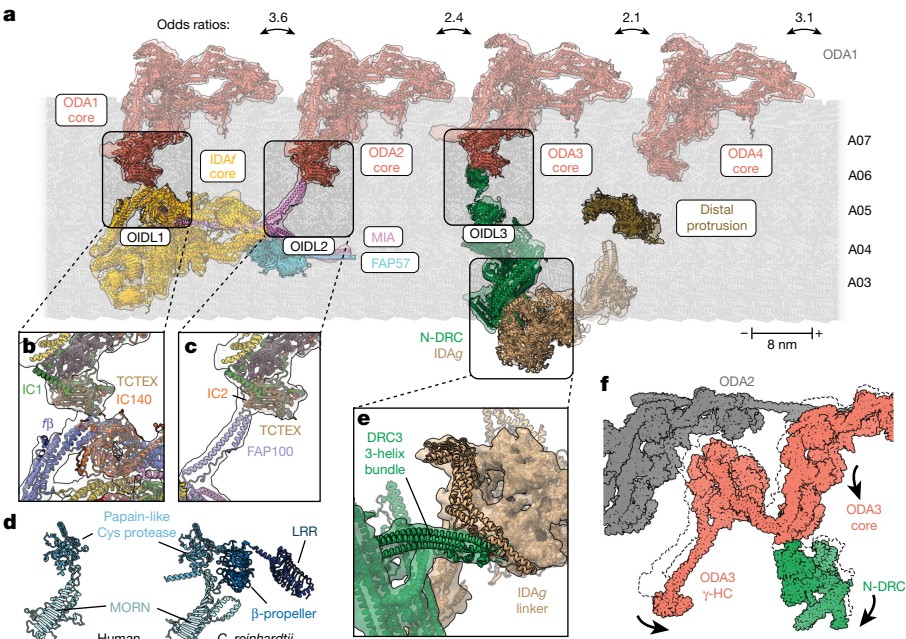

**Fig. 3 | Regulatory complexes modulate inter-dynein interactions.**
**a**, Selected atomic models and their cryo-EM densities showing the longitudinal arrangement of ODA1–ODA4 and their adjacent complexes on the axoneme. Each ODA exists in open and closed conformations, as revealed by 3D classification of the cryo-EM dataset. Odds ratios for the co-occurrence of open and closed states between ODA pairs are shown above the bidirectional arrows. Odds ratios were calculated using 160,444 particles for ODA1, 162,179 particles for ODA2, 214,203 particles for ODA3 and 177,762 particles for ODA4. Scale bar, 8 nm (the length of a tubulin heterodimer) with microtubule polarity indicated. **b**, OIDL1 between the TCTEX of ODA1 and IDA*f* subunits IC140 and *f*β. **c**, OIDL2 between the TCTEX of ODA2 and MIA subunit FAP100. **d**, Models of human (left) and *C. reinhardtii* (right) DRC7 with the domains coloured in different shades of blue. Human DRC7 lacks the β-propeller and LRR domains that mediate OIDL3 in *C. reinhardtii*. MORN, membrane occupation and recognition nexus. **e**, Interaction between the three-helix bundle of DRC3 and the linker of IDA*g*. **f**, Overlay of the two major conformational states of N-DRC. Arrows denote movement from the closed to the open state.

neighbour into an open state (Supplementary Videos 2 and 3) and also favours separation between the γ- and β-HC tails near the OIDL. We considered that breaking inter-ODA contacts in one complex may send a longitudinal signal that favours the breaking of inter-ODA contacts in adjacent ODA complexes. In support of this hypothesis, we calculated odds ratios between 2.1 and 3.6 for all neighbouring ODA complexes (Fig. 3a, top), indicating that ODA conformation is influenced by the neighbouring ODA. The open and closed states of ODA1 and ODA2 are highly correlated, with an odds ratio of 3.6 (Fig. 3a), probably through their connections to IDA*f*. However, the conformations of ODA3 and ODA4 with their adjacent ODA complexes are also correlated without a mechanoregulator physically linking them (Fig. 3a). We therefore propose that longitudinal signals caused by the breaking of inter-ODA contacts may activate clusters of ODAs during the propagation of the waveform from the proximal to the distal end of the cilium.

## The modular repeat of the human axoneme

To investigate species-specific differences between axonemes, we used our cryo-EM approach to determine a structure and atomic model of the 96-nm modular repeat of human DMT isolated from ciliated 2D organoids resembling the respiratory epithelium (Fig. 4a,b and Supplementary Figs. 13 and 14). The model clarifies the composition of human RS1 and RS2 (Supplementary Tables 3 and 4), revealing simpler spoke heads, necks and bases (in the case of RS1) than in *C. reinhardtii* (Fig. 4c). The simpler architectures of RS1 and RS2 may be compensated by a full-length RS3, which we show is built on a base of CFAP61, CFAP91 and CFAP251 (Fig. 4c and Extended Data Fig. 6a). The homologues of this triumvirate were previously assigned to the

calmodulin- and spoke-associated complex[35,36], which we show form the stubby, headless RS3 stand-in (RS3S) in *C. reinhardtii* (Extended Data Fig. 6b,c). Further investigation is required to identify the proteins that bind CFAP61, CFAP91 and CFAP251 to form the rest of RS3 in humans.

Our model demonstrates that human IDA*f* and the ODA are even more structurally similar than in *C. reinhardtii* because there are fewer additional subunits, including the α-HC and equivalents of LC5 and FAP120. One difference is the direct contact between the TCTEX heterodimer of human IDA*f* and the linker of the IDA*a* motor domain, potentially providing an additional signal-transduction pathway between axonemal dyneins. Unlike ODAs, which have structurally distinct docking complexes in mammals and algae[5,37], the docking machinery for IDA*f* is highly conserved. The only difference is the absence of a MOT7 equivalent at the centre of the T/TH stellate structure, even though the human genome encodes a potential homologue (TEX47). The site adjacent to ODA4, which is occupied by the distal protrusion in the *C. reinhardtii* axoneme, is vacant (Extended Data Fig. 5e,f), as expected from cryo-ET[32] and the restricted taxonomic distribution of FAP78 homologues (Extended Data Fig. 5g).

## Species-dependent IDA organization

An unexpected difference between the axonemes of *C. reinhardtii* and humans is the composition and arrangement of the six single-headed IDAs (Fig. 5a,b). In *C. reinhardtii*, the molecular identity of each HC that forms the IDAs was previously known[38,39], but in humans, the identities of only IDA*f* (DNAH2 and DNAH10), IDA*d* (DNAH1) and IDA*g* (DNAH6) had been confidently deduced on the basis of sequence conservation[40]. By distinguishing structurally distinct HC tails from the cryo-EM

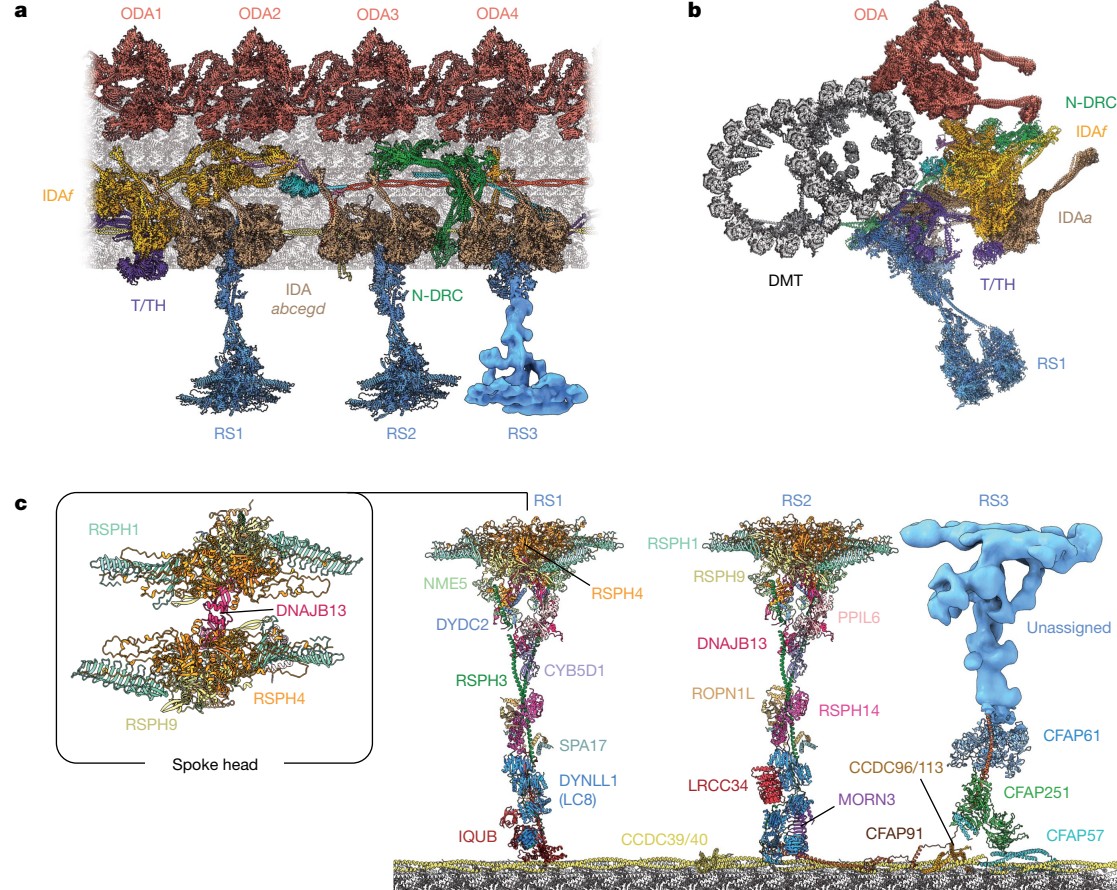

**Fig. 4 | Atomic model of the 96-nm repeat of a human axoneme.**
**a**, Longitudinal section of the 96-nm repeat of a human respiratory-cilia DMT.
**b**, Cross-section of a human DMT. **c**, Atomic model of human RSs coloured by
density map in **a** and **c**.

subunit. The inset shows the symmetric head of RS1 viewed from above. Only
the composition of the base of RS3 is known; the rest of the complex is shown as
a density map in **a** and **c**.

density, we have now confirmed those HC assignments[40] and also discovered that IDA*a* is DNAH12, IDA*c* is DNAH3, and IDA*b* and IDA*e* are both DNAH7 (Fig. 5c,d).

IDA*a*, IDA*c*, IDA*d* and IDA*g* are similar in both organisms. IDA*a* and IDA*c* both have actin and p28/DNALI1 subunits and are directly associated with the microtubule-bound bases of RS1 and RS2, respectively. IDA*d* and IDA*g* are unusual among single-headed IDAs because their N-terminal tails and auxiliary subunits are amalgamated into a single complex that associates with RS3 and ZMYND12/TTC29 complex in humans or the equivalent RS3S and p38/p44 in *C. reinhardtii* (Fig. 5b,e). Thus, the most-conserved IDAs are those that associate with RSs. By contrast, human IDA*b* and IDA*e* differ from their *C. reinhardtii* counterparts in that they have HCs encoded by a single gene, interact with a DNALI1 homodimer instead of centrin and dock onto different sites on the DMT surface. In *C. reinhardtii*, IDA*b* binds at the distal end of IDA*f* on top of the L-shaped molecular ruler (a dimer of the FAP189 family), and IDA*e* binds beside the bulb of the N-DRC over protofilament A04. In humans, IDA*b* and IDA*e* are positioned in line with, and closer to, IDA*a* and IDA*c* (Fig. 5b), where they interact directly with the tubulin surface of protofilament A03 through the DNALI1 subunit (Fig. 5f), rather than with a regulatory complex or docking factor. Despite the different binding location, human IDA*e* still interacts with the N-DRC bulb, which has uncoupled from the baseplate and shifted distally (Extended Data Fig. 4f,g). IDA*e* is lost in *C. reinhardtii* mutants that lack DRC1 or DRC2 (ref. 41), but whether the N-DRC is required for the docking of human IDA*e* is unknown.

## Cryo-EM analysis of mutant axonemes

We then used our cryo-EM approach to investigate how PCD-associated mutations affect human axonemal structure. We prepared axonemes from ciliated organoids derived from four individuals diagnosed with PCD using standard diagnostic criteria (Methods and Extended Data Table 2) and carrying sequence-verified homozygous variants (Extended Data Fig. 7a) selected to disrupt either the 24-nm repeat (by targeting the ODAD1 subunit of the ODA-DC) or the 96-nm repeat (by targeting the CCDC39–CCDC40 (CCDC39/40) coiled coil). All four mutations have been either shown[42] or predicted to create null alleles owing to frameshift or nonsense-mediated degradation, leading to total protein loss. The positions of the affected proteins in the wild-type (control) axoneme are depicted in Fig. 6c.

Each mutation caused immotile cilia (more than 90% for *ODAD1* and between 50% and 60% for *CCDC39* and *CCDC40*), with any beating cilia displaying low beat frequency and amplitude (Fig. 6a) and abnormal waveforms (Fig. 6b, Extended Data Fig. 7b and Supplementary Video 4). Clinical transmission electron microscopy (TEM) analysis showed that *ODAD1* mutations caused ODA loss, whereas *CCDC39* and *CCDC40* mutations caused disorganized cross-sections with IDA-deficient DMTs and microtubule transposition (Extended Data Fig. 7c). To better understand how these mutations affect axonemal structure beyond what can be visualized by thin-section TEM, we used cryo-EM to generate structures of their DMTs. For the two *ODAD1* mutants (from patients ID01 and ID02) we could determine structures of only their 8-nm repeats owing to the limited recovery of cilia (Fig. 6d). Comparison

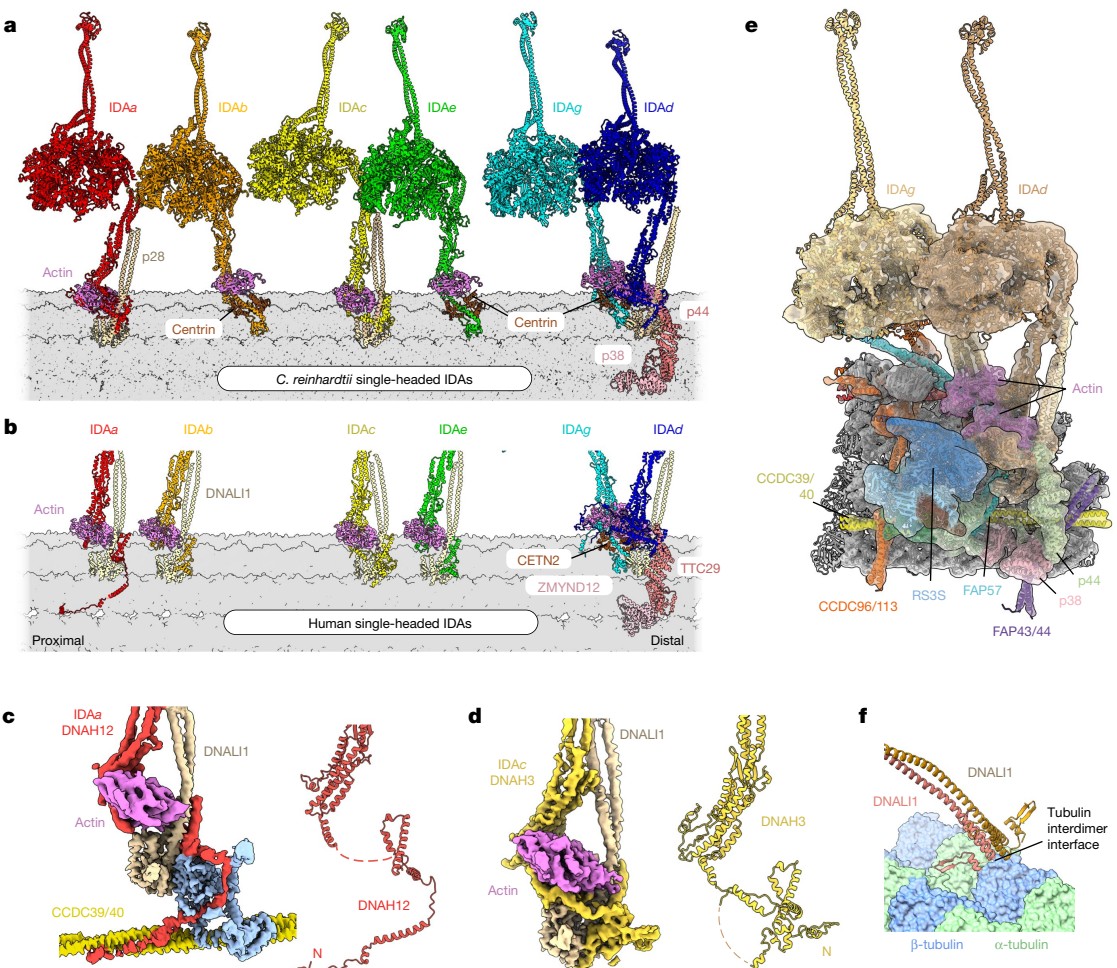

**Fig. 5 | Species-specific compositions and positions of single-headed IDAs. a**, Atomic model of the six single-headed IDAs as they appear on the *C. reinhardtii* DMT. Each IDA is associated with actin and either centrin or a p28 homodimer; p38 and p44 are specific docking factors for IDAd. **b**, Cropped image showing the DMT-bound bases of the human single-headed IDAs. Compared with those of *C. reinhardtii*, IDAb and IDAe have different subunit compositions and docking positions. **c,d**, Cryo-EM density of the dynein tail domains was used to identify the HC of IDAa as DNAH12 (**c**) and that of IDAc as DNAH3 (**d**). **e**, Cryo-EM map and model showing the molecular environment of IDAg and IDAd and its connection with RS3S in *C. reinhardtii*. A similar arrangement is found at the base of human RS3. **f**, The DNALI1 homodimer of human IDAe interacts directly with the tubulin interdimer interface. Similar interactions with tubulin are observed for human IDAa, IDAb and IDAc and their equivalents in *C. reinhardtii* (through the equivalent p28 homodimer).

with a wild-type DMT structure reconstructed from a similar number of particles to the same resolution and periodicity demonstrated a complete loss of the ODA-DC. Internal MIP structures and the external 96-nm repeat seemed to be unaffected. The structures of the DMTs from the *CCDC39* and *CCDC40* mutants (from patients ID03 and ID04) could be resolved to higher resolution and 48-nm periodicity, the maximum possible, given the complete absence of any recognizable 96-nm repeat, including the CCDC39/40 coiled coil (Fig. 6e). In the absence of regular 96-nm periodicity, RSs are imported into the axoneme as usual but bind DMTs only sporadically (Extended Data Fig. 7d). In both structures, ODA-DCs and the 24-nm repeat were unaffected but minor defects in the composition of the MIP tektin network were observed.

Collectively, these DMT structures from patients with PCD demonstrate that the loss of docking factors can lead to the complete loss of external periodicities, even when the components (such as RSs) are present in the cilium. The studies also reveal that the 24- and 96-nm external periodicities of the axoneme are independent, as one can be lost without affecting the other, which is consistent with previous observations[43] in *C. reinhardtii*. Furthermore, the loss of external structures had only minor effects on MIP organization, indicating that the establishment of the internal 48-nm repeat is independent of the 96-nm repeat.

## Discussion

Our ability to create atomic models of the 96-nm axonemal repeats from two different organisms (*C. reinhardtii* and *Homo sapiens*) demonstrates the strength of combining cryo-EM with AI-based structure prediction for building models of complicated, dynamic cellular machinery. The resulting atomic models reveal that although the axoneme is structurally conserved, there are species-specific differences present in every axonemal complex. The similarities reveal fundamental mechanisms involved in the periodic docking of axonemal complexes onto microtubules and the engagement and manipulation of dyneins by mechanoregulatory sensors. On the other hand, the differences reveal species-specific factors and regulatory mechanisms that fine-tune ciliary waveforms.

### Structural diversity of axonemal dyneins

This study comprehensively resolves the structural diversity of axonemal dyneins. We show that multi-headed ODAs and IDAf share many structural features, consistent with their proposed evolution by duplication[40]. MIA and the coiled coil of the ODA-DC may have similar evolutionary origins given their apparent functional equivalence as

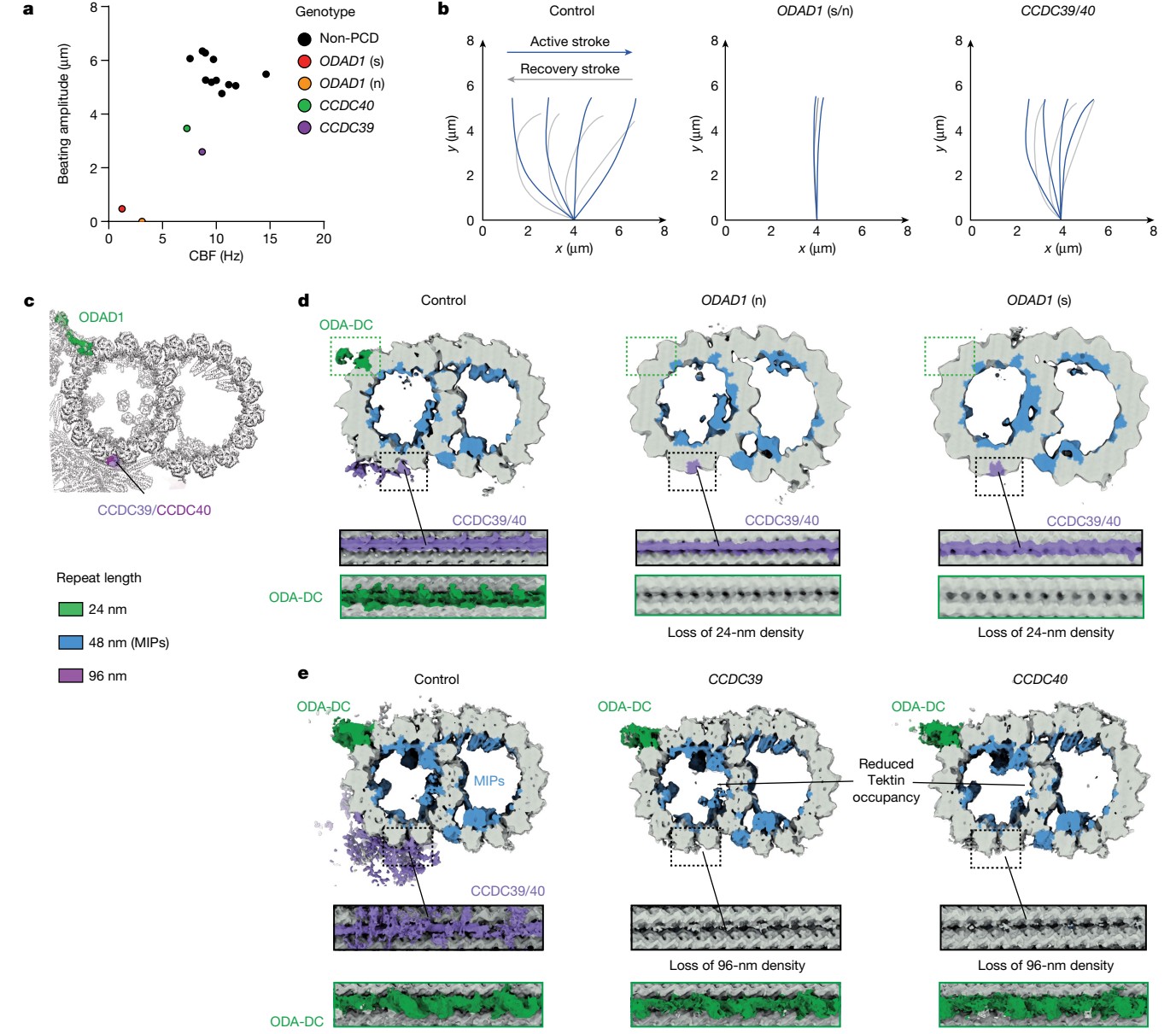

**Fig. 6 | Structures of DMTs from individuals with PCD. a**, Beating amplitude is plotted against ciliary beat frequency (CBF) for primary brushing samples from human nasal epithelia for 11 people without PCD and patients with PCD carrying biallelic *ODAD1* (splice site (s) and nonsense (n)), *CCDC39* and *CCDC40* mutations. **b**, Schematic models of the corresponding ciliary waveforms observed in primary nasal brushings of patients with PCD and people without PCD (control). **c**, Cross-section of the atomic model of the human 96-nm DMT repeat depicting the position of ODAD1 and the CCDC39/40 coiled coil. The proteins are coloured according to their periodicity in the axoneme. **d**,**e**, Cross and longitudinal sections through cryo-EM maps reconstructed from DMTs isolated from pseudostratified respiratory epithelia cultured from either a healthy individual (control) or nasal brushings from patients with PCD. Map densities are coloured according to the repeat length with tubulin shown in grey. **d**, Mutations in *ODAD1* cause complete loss of the ODA-DC (green), which repeats with 24-nm periodicity. **e**, Mutations in *CCDC39* and *CCDC40* cause complete loss of the 96-nm repeat (mauve). Apart from tektins, which had reduced occupancy, most MIPs (blue) are unaffected. Control maps are reconstructed from similar numbers of particles at a similar resolution and periodicity to the mutant maps for direct comparison.

heterodimeric dynein-docking factors. The differences in the subunit composition and binding sites of the single-headed axonemal dyneins highlight the role of these IDAs in the diversification of axonemal structure and the species-specific fine-tuning of ciliary motility. We show that their divergent N termini are responsible for their localization to specific axonemal substructures, explaining why IDAs are not functionally interchangeable and cannot be substituted by other homologues[3,39]. Our structural information will aid the investigation of dynein HCs in other species because dyneins with structurally conserved N termini are expected to bind to similar positions in the axoneme.

## Multipartite docking mechanisms

Our findings provide insights into the complex mechanisms that generate and maintain periodicities across the axoneme. In contrast to previous studies that reinforced the idea that axonemal complexes have single dedicated docking machineries (such as the ODA-DC[5,37]), we demonstrate that multipartite docking mechanisms exist for other axonemal complexes, including IDA*f* (Fig. 2a,b). These docking factors can span large distances and contribute to the binding of multiple axonemal complexes. For example, two of the IDA*f* docking factors,

FAP57 and the C-terminal FAP43/44 coiled coil of T/TH, also function in the multipartite docking of IDA*g* and IDA*d* (Fig. 2b, Fig. 5e, and Extended Data Fig. 6d), which explains the partial loss of IDA*g* and IDA*d* in *fap57* mutants[23]. The physical interlocking of shared docking factors seems to provide a robust mechanism to maintain the precise organization and periodicity of the 96-nm repeat over many micrometres. This type of interlocking is also observed for CFAP91, which begins at the base of RS2, interacts with the N-DRC[14] and then links the RS3 subunits CFAP251 and CFAP61 (Fig. 4c).

## Insights into axonemal asymmetry

In *C. reinhardtii*, variations in the identity of the coiled-coil docking factors may provide a mechanism for establishing the longitudinal and radial asymmetry of axonemes. We observed that two surface-bound coiled coils, FAP57 and the L-shaped FAP189, have paralogues in *C. reinhardtii* (Extended Data Fig. 8). Our maps probably represent averages of these paralogues in which the quality of the density is determined by sequence conservation among paralogues (Extended Data Fig. 8b–g). We propose that these paralogues are distributed asymmetrically across the axoneme to regulate IDA composition and function. Consistent with this hypothesis, loss of FAP57 results in partial loss of IDA*g* and IDA*d* from only DMT1 and DMT5–DMT9 (ref. 23). The functional importance of different paralogues to *C. reinhardtii* motility is seen in the *mbo2* mutant strain that lacks MBO2 (a paralogue of FAP189) and the ability to convert between asymmetrical and symmetrical waveforms[44]. A similar mechanism for establishing asymmetry may operate in humans, who have two orthologues of FAP189 (CFAP58 and CCDC146).

## A mechanoregulation mechanism for ODA regulation

By mining our particle dataset for conformational changes, we present a structure-based model for how IDA*f* and N-DRC could regulate ODA activity. In this model, the extensive head-to-tail inter-ODA contacts in the post-powerstroke state constrain dynein conformation, such that this state must be relaxed before ATP turnover and transition to the pre-powerstroke state[6], which has a remodelled set of inter-ODA interactions[45]. Local deformation of the inter-ODA contacts could therefore act to regulate ODA activity by lowering the energy barrier for ATP hydrolysis and facilitating the transition to the pre-powerstroke state, whereas inter-ODA stabilization would be inhibitory. Genetic studies in *C. reinhardtii* favour inhibition. N-DRC and IDA*f* mutations partly suppress the 'paralysed flagella' phenotype caused by lack of RSs or the central apparatus[46,47], and loss of IDA*f* restores the velocity of ODA-driven microtubule disintegration in mutants that lack the central apparatus[48]. However, high-resolution cryo-ET of rapidly frozen, actively beating cilia will be needed to correlate conformational changes in the N-DRC and IDA*f* with dynein activity to determine whether inhibition is relieved by microtubule bending. Our atomic models will help in the interpretation of these cryo-ET studies.

The observed correlation between neighbouring ODA conformations (Fig. 3a) suggests a mechanism for propagating longitudinal signals through the ODA array to create a proximal-to-distal waveform. Crucially, this signal is not dependent on each ODA being directly connected to a mechanoregulator. Connections that are not conserved outside algae, such as the DRC7-mediated OIDL3 (Fig. 3d), may represent a gain of function that provides a supplemental or redundant mechanism of regulating the ciliary waveform.

As well as regulating ODA behaviour, the N-DRC seems to modulate IDA*g* activity through its conserved interaction with DRC3 in *C. reinhardtii* (Fig. 3e) and humans (Extended Data Fig. 4h). As the N-DRC tilts, the IDA*g* linker and DRC3 form a closer connection (Supplementary Video 3), which could constrain IDA*g* conformation or activity. The functional importance of this interaction can be seen in *C. reinhardtii drc3* mutants, which have an altered flagellar waveform despite having an otherwise fully assembled N-DRC[34], and the formation of hydrocephalus and male infertility in a point mutation of the mouse *drc3*

homologue[49]. Overall, our results support a mechanism whereby the N-DRC transmits a mechanical signal through its electrostatic interaction with the neighbouring DMT (Fig. 2e) through a conformational change to ODA3 and IDA*g*.

## Cryo-EM as a clinical tool

Our human DMT model contains around 25 proteins linked to PCD and 20 others associated with either PCD-like phenotypes or male infertility (Supplementary Table 5). The model therefore represents a valuable clinical resource for evaluating the likely effect and pathogenicity of disease mutations and identifying new disease candidates for unsolved PCD cohorts. We recommend expanding clinical genetic screening for PCD and male infertility to include all the genes that encode structurally verified proteins of the axoneme. We anticipate that this expansion may help to account for the 20–25% of PCD cases that cannot currently be explained genetically.

Furthermore, our ability to resolve cryo-EM structures of DMTs from individuals provides an experimental approach to examine the impact of PCD-causing mutations on axonemal structure. We have shown that the translational potential of this approach is particularly powerful when combined with molecular genetics. Compared with cross-sectional TEM and even 3D tomography, the higher resolution provided by single-particle cryo-EM allows the detection of minor structural changes, such as perturbation of the tektin network. Cryo-EM could therefore serve as a useful tool to analyse the 30% of PCD cases in which the axonemal ultrastructure seems to be normal by clinical TEM[50]. Alternatively, by applying our approach to detect variations in cilia across different tissues, we may gain new insights into the molecular basis for the varying susceptibility to dysfunction caused by mutations in specific genes.

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

## Methods

### Human subjects

Peripheral blood samples for genetic screening and respiratory epithelial cells, collected from the inferior nasal turbinate by nasal-scrape brushing, were obtained from affected individuals and healthy volunteers under ethical approval granted through the Health Research Authority London Bloomsbury Research Ethics Committee (REC 08/H0713/82; IRAS 103488) and Living Airway Biobank, administered through the UCL Great Ormond Street Institute of Child Health (REC 19/NW/0171, IRAS 261511, Health Research Authority North West Liverpool East Research Ethics Committee). Informed written consent was obtained from all participants before enrolment in the study. Affected individuals were diagnosed with PCD according to European Respiratory Society (ERS) diagnostic guidelines[51].

### Human genetic analysis

Genetic screening in patients with PCD was performed as previously described[42] for individual ID01 and by using a targeted next-generation sequencing panel[52] for individuals ID02, ID03 and ID04. A definitive genetic diagnosis was obtained according to ERS diagnostic guidelines[51], involving the identification of biallelic pathogenic variants (found in a homozygous state in each case) in single PCD genes consistent for each defect observed by TEM.

The first individual (ID01) is homozygous for a previously reported c.853G>A substitution in *ODAD1* that causes an intronic insertion that ultimately results in a downstream nonsense codon (p.Ala248Thrfs52*)[42]. The second individual (ID02) carries a homozygous c.448C>T substitution that introduces a premature stop codon in *ODAD1* (p.Arg150*) and has previously been associated with PCD[53]. The third individual (ID03) has a homozygous exon 1–3 deletion in *CCDC40*, and the fourth individual (ID04) has a homozygous 5-base-pair deletion (c.1964_1968delCTCTT) in *CCDC39* that introduces a premature stop codon through a frameshift (p.Glu655Glyfs*23).

### Clinical TEM

TEM of nasal epithelial cells was performed at the University of Leicester, UK. Cells obtained by nasal brushing were fixed in 2.5% glutaraldehyde in either Sorenson's phosphate or 0.05 M sodium cacodylate buffer for more than 24 h. Samples were subsequently incubated in 1% osmium tetroxide before embedding in 2% agar. Samples were dehydrated using an alcohol series that included increasing concentrations of ethanol/methanol and propylene oxide before embedding in epoxy resin. Ultrathin sections were cut and stained with uranyl acetate and Reynold's lead citrate. Images were acquired using a room-temperature JEOL 1400+ transmission electron microscope. Representative examples are provided in Extended Data Fig. 7c.

### Cilia beat analysis

Videos of ciliated respiratory epithelium were acquired by high-speed video microscopy and analysed using a program designed in MATLAB R2022a (MathWorks) based on previous publications[54,55]. In brief, positions at the base and tips of cilia at the beginning and end of an active phase of a beat cycle were recorded. Ciliary beat frequency was calculated by fast Fourier transform and video-kymography. This analysis allowed the calculation of ciliary beat pattern parameters, such as cilium length, angle of beating and beating amplitude. Ciliary beat amplitude per second (beating amplitude × ciliary beat frequency) has previously been described as the best approach to discern between individuals with and without PCD[55]. We analysed 104 videos from 11 individuals without PCD as a control.

### Epithelial disruption scores

Epithelial disruption scores were calculated from videos of nasal-scrape brushings as described previously[56]. In brief, epithelial edges were identified in each sample, investigated for protrusions or abnormalities, and scored from 1 to 4, where 1 indicates a normal ciliated edge, 2 indicates a ciliated edge with minor projections, 3 indicates a ciliated edge with major projections, and 4 is for single, unattached, ciliated cells.

### Respiratory epithelial culture and cilia isolation

Cells obtained from nasal-scrape brushings (of both a healthy individual and four individuals with PCD) were seeded into collagen-coated wells (PureCol, Sigma-Aldrich, 5006-15MG) containing PneumaCult-Ex Plus medium (STEMCELL Technologies, 05040). The cells were passaged twice: once into a T25 flask and then onto 12 Transwell inserts, each with a surface area of 1.12 cm² (Corning, CLS3460) at a density of around 0.5–1.0 million cells per insert. Cells were cultured in PneumaCult-Ex Plus medium until confluent, at which time the medium in the basal chamber was replaced with PneumaCult-ALI medium (STEMCELL Technologies, 05001) and the apical surface was exposed to provide an air–liquid interface (ALI). All media contained Primocin (InvivoGen, ant-pm-05) and penicillin–streptomycin (Thermo Fisher Scientific, 15070063) to prevent the growth of bacteria. Ciliation was observed 4–6 weeks after transition to the ALI. During differentiation, the basolateral medium was refreshed every 2–3 days and the apical surfaces were washed with PBS to remove mucus.

Before deciliation, differentiated ALI cultures were washed twice with PBS for 5 min to remove cell debris and mucus. Ice-cold PBS was then added to both compartments of the culture dish and the dish was placed on ice. After 5 min, the PBS solution was removed and 50 µl ice-cold deciliation buffer (10 mM Tris pH 7.5, 50 mM NaCl, 10 mM CaCl₂, 1 mM EDTA, 0.1% Triton X-100, 7 mM β-mercaptoethanol, 1% protease inhibitor cocktail; Sigma-Aldrich, P8340) was added to the cells of each well. After incubation for 2 min without shaking, the cilia-containing solution was transferred to a microcentrifuge tube. This process was repeated six times to ensure that all dissociated cilia were collected. Cellular debris and mucus were removed by centrifugation at 1,000*g* for 1 min at 4 °C. Axonemes were collected by centrifugation at 15,000*g* for 5 min at 4 °C. The axonemal pellet was resuspended in buffer (30 mM HEPES pH 7.3, 1 mM EGTA, 5 mM MgSO₄, 0.1 mM EDTA, 25 mM NaCl, 1 mM dithiothreitol, 1% protease inhibitor cocktail), flash-frozen in liquid nitrogen and stored at −80 °C until use.

### Sample preparation for cryo-EM

**Splayed *C. reinhardtii* axonemes.** *C. reinhardtii* (CC-1690) algae were cultured in standard Tris acetate phosphate medium at room temperature in 12 h:12 h light:dark cycles. Flagella were extracted from *C. reinhardtii* using the dibucaine method[57], as described previously[5]. After removing the cell bodies, the detached flagella in the supernatant were collected by centrifugation at 3,000*g* for 15 min. Freshly isolated flagella were splayed into DMTs as described previously[5]. In brief, flagellar membranes were first removed in fresh HMDEKP (30 mM HEPES, 25 mM KCl, 5 mM MgSO₄, 0.5 mM EGTA, Protease Arrest (G-Biosciences)) by adding 1% NP-40 at 4 °C while rotating for 30 min. After washing away the detergent and other soluble proteins, axonemes were splayed at a final concentration of 0.5 mg ml⁻¹ by treatment with 10 mM Mg²⁺ATP²⁻ and 750 µM CaCl₂ in HMDEKP while being rotated at room temperature for 1 h. Splayed axonemes were concentrated by centrifugation at 2,500*g* and immediately used to prepare cryo-EM grids. Then 2.5 µl of splayed axoneme sample at 16–26 mg ml⁻¹ was applied to glow-discharged C-Flat 1.2/1.3-4Cu grids (Protochips) inside a Vitrobot Mark IV (Thermo Fisher Scientific) kept at 100% humidity. The cryo-EM samples were then incubated 10 s, blotted for 10 s and plunged into liquid ethane cooled by liquid nitrogen.

**Human respiratory-cilia axonemes.** Human axonemes (3 µl) with an absorbance reading at 280 nm of 5.8 were applied to glow-discharged Quantifoil holey carbon grids (R2/1, copper, 400 mesh, Quantifoil

Micro Tools, Q410CR1). The grids were blotted for 10–11 s with a blot force of 10 in 100% humidity before being plunged into liquid ethane using a Vitrobot Mark IV (Thermo Fisher Scientific).

## Cryo-EM data collection and processing for splayed *C. reinhardtii* axonemes

A total of 36,918 micrographs of splayed *C. reinhardtii* axonemes were collected using SerialEM as described[58] on a Titan Krios microscope (Thermo Fisher Scientific) under conditions identical to our previous study[5]. Microscope settings are summarized in Extended Data Table 1.

All image processing was done using RELION-3.1 (ref. 59) or RELION-4.0 (ref. 60) unless otherwise stated. The dose-fractionated image stacks were aligned and dose-weighted using MotionCor2 software[61]. CTFFIND4 was used to estimate the parameters of the contrast transfer function (CTF)[62]. A total of 31,275 micrographs survived quality control on the basis of the absence of ice contamination, the quality of the Thon rings and the presence of splayed axonemes in the micrographs. Microtubules (including DMTs and singlet microtubules) were manually picked on each micrograph by defining their start and end coordinates in RELION. Overlapping microtubules were avoided. The selected microtubules were computationally divided into overlapping boxes (200 × 200 pixels, bin 2) with an 82-Å non-overlapping region (step size) between adjacent boxes, corresponding to the approximate length of the tubulin α/β-heterodimer. A total of 8,546,747 8-nm particles were extracted and aligned by multiple rounds of 2D classification, with the 106,857 particles that fell into poorly defined classes being discarded. The remaining 8,439,890 particles were analysed using both a DMT-centred approach[13] and an ODA-centred approach[5] (Supplementary Figs. 1–3). In the DMT-centric method, the 8-nm particles underwent 3D refinement and classification to select 2,058,898 particles with well-resolved DMT density. These particles were re-extracted without downscaling in 600-pixel boxes, 3D-refined, CTF-refined, polished and CTF-refined again. In the ODA-centric method, the 8-nm particles underwent 3D refinement and classification to yield three ODA-bound classes. These three classes corresponded to ODA centred in the map and two shifted by ±8 nm from the centre. The off-centre classes were re-extracted to centre the ODA in the map to generate a '24-nm particle' set. To maximize particle number, we extracted new particles 24 nm and 48 nm from this set. A second round of 3D refinement and classification yielded 1,028,815 24-nm particles with a centred ODA. These particles were then re-extracted without downscaling in 600-pixel boxes, 3D-refined, CTF-refined, polished and CTF-refined for a second time.

For both the DMT- and ODA-centric approaches, maps of the 48-nm repeat were obtained by 3D classification using a mask on the lumen of A-tubule protofilaments A08–A13 where MIPs have 48-nm periodicity[13]. For the DMT-centric approach, this yielded six different registries of the 48-nm repeat, and for the ODA-centric approach, two registries from the previously determined 24-nm repeat maps were identified. We then obtained 96-nm maps by a second round of 3D classification using a custom mask over the external axonemal complexes, including RSs, the N-DRC baseplate and IDA*f* to yield a total of 12 different registries from the DMT-centric approach, and 4 different registries from the ODA-centric approach.

To overcome conformational heterogeneity and loss of individual complexes during sample preparation, each axonemal complex was analysed by 3D classification and refinement following the subtraction of signal outside the area of interest (Supplementary Figs. 4–9). Where appropriate, maps were obtained by combining subtracted particles from different registries and from both the DMT- and ODA-centric approaches. When combining particles, strict controls were used to prevent particle duplication.

Focused maps of individual axonemal complexes were positioned on the DMT using unmasked box 600 reference maps from 3D refinement containing IDAs, ODAs and the N-DRC. Because of the low resolution, the motor domains of IDA*b* and IDA*e* are copies of IDA*g*, the other centrin-containing IDA. The maps were then combined into a single composite map that covered a 96-nm section of the DMT using the 'vop maximum' command in ChimeraX[63]. The initial composite map was then cropped using the subregion selection tool of Chimera to remove empty regions and reduce the file size.

## Cryo-EM data collection and processing for human respiratory DMTs

A total of 37,071 micrographs of human respiratory DMTs were acquired using a Titan Krios microscope (Thermo Fisher Scientific) under conditions identical to our previous study[64]. Microscope settings are summarized in Extended Data Table 1. The videos were motion- and CTF-corrected using MotionCor2 (ref. 61) and CTFFIND4 (ref. 62), respectively. 16,933 micrographs survived quality control. Microtubule start and end points were selected from the micrographs using RELION manual picking. 2,582,833 8-nm particles were extracted from the micrographs in 512-pixel boxes, downscaled to 256-pixel boxes to accelerate computation. 2D classification was done to check image quality, not to exclude particles.

A 15-Å-resolution map of the bovine respiratory DMT (EMD-24664)[37] was used as an initial reference for 3D refinement, followed by 3D classification without image alignment to exclude bad particles. To increase the number of retained particles, we repeated the process and merged all the good particles while excluding duplicates. In total, 1,903,665 8-nm particles were retained and re-extracted without binning. Then 16-nm particles were generated from the 8-nm particles using 3D classification with a cylindrical mask on 16-nm repeating MIPs near the inner junction. Subsequently 48-nm particles were generated from the 16-nm particles using 3D classification with a cylindrical mask on 48-nm repeating MIPs near the seam of the A tubule. Finally, 96-nm particles were generated from the 48-nm particles using 3D classification with a cylindrical mask on 96-nm repeating axonemal proteins on protofilaments A02–A03. This resulted in two parts of the 96-nm repeat with 90,482 and 87,789 particles. Local refinements using 60 different cylindrical masks, each corresponding to a 16-nm longitudinal section of 2 or 3 protofilaments, were then performed to improve the resolution of the protofilaments and closely associated microtubule-binding proteins. The resolution within each mask improved to 3.3–3.8 Å. The map quality of the external axonemal complexes was improved using particle re-centring, 3D classification, mask-focused local refinement and multi-body refinement. The resulting maps and their nominal resolutions (based on the FSC = 0.143 criterion) are listed in Supplementary Fig. 14. Only 13,340 ODA-containing particles were observed, which prevented analysis of different conformational states, as done for our *C. reinhardtii* dataset. Maps were postprocessed using Phenix_autosharpen and merged in ChimeraX to generate a composite map. A schematic of the processing scheme is provided in Supplementary Fig. 13.

Cryo-EM data of respiratory DMTs from human patients with mutations in *CCDC39* and *CCDC40* were collected on a Titan Krios microscope under conditions identical to those for the wild type, yielding 3,423 and 3,483 micrographs, respectively. Cryo-EM data of respiratory DMTs from human patients with *ODAD1* mutations were collected on a Talos Arctica microscope (Thermo Fisher Scientific) equipped with a K3 detector (Gatan), yielding 56 and 146 micrographs of the *ODAD1* nonsense mutant and the splice mutant, respectively. Data were collected with a defocus range between −1.0 μm and −2.2 μm and at a nominal magnification of ×36,000, yielding a pixel size of 1.1 Å. Each micrograph was fractionated into 50 video frames with a total dose of 52.562 electrons per Å$^2$. Data processing followed the scheme established for the wild-type dataset (Supplementary Fig. 13). For data processing of the *CCDC39* and *CCDC40* mutants, around 280,000 8-nm particles were randomly selected and used to generate 48-nm maps. The same number of wild-type particles were processed in the same way to generate a

matched control. The three 48-nm maps were lowpass filtered to 8 Å for comparison in Fig. 6e. For data processing of the *ODAD1* mutants, 12,884 and 35,541 8-nm particles were extracted for the *ODAD1* nonsense mutant and splice mutant, respectively. Only 8-nm maps were generated because of the limited particle numbers. *ODAD1* mutant maps and an 8-nm wild-type map (from the above-mentioned 280,000 particles) were lowpass filtered to 12 Å for comparison in Fig. 6d.

## Model building and refinement

Maps of the *C. reinhardtii* DMT were interpreted by combining previous structures, AI-guided structural predictions[65], cryo-ET studies[3,24,39,41,66–68] and de novo modelling using Coot[69]. Previous structures were used to model the DMT and its associated MIPs (PDB: 6U42)[13], RS1 and RS2 (PDB: 7JTS, 7JTK and 7JU4)[14], the N-DRC baseplate (PDB: 7JU4)[14] and the ODA with its docking complex (PDB: 7KZM and 7KZO)[5] and LC1 subunit (PDB: 6L4P)[70]. Models of the C1 (PDB: 7SQC)[15] and C2 (PDB: 7SOM)[15] microtubule were used to make Fig. 1b. Structure predictions of other axonemal proteins were obtained using AlphaFold2 (ref. 71) using primary sequences from Phytozome v.13 and UniProt. Homology models of the dynein motor domains (including the linker, AAA+ modules, stalk and microtubule-binding domain) were generated in the post-powerstroke state using SWISS-MODEL[72], as they were too large to be modelled by AlphaFold2. Protein–protein interactions were modelled using AlphaFold2 multimer[73]. Examples of subcomplexes modelled using AlphaFold2 multimer are shown in Supplementary Figs. 10–12. Axonemal complexes were assembled from modelled proteins and subcomplexes by positioning them in the cryo-EM maps using ChimeraX[63] or Coot[69]. The subtomogram average of the 96-nm repeat from *C. reinhardtii* (EMD-6872)[24] was used to help position the motor domains of IDA*b* and IDA*e*. Multiple rounds of AlphaFold2 multimer prediction were used to improve the interfaces between neighbouring proteins. Finally, all chains were merged into a single PDB file using Coot and given a unique chain ID. Proteins identified in this study are provided in Supplementary Table 1 and the rationale for their placement is given in Supplementary Table 2.

Our splaying method of sample preparation causes axonemal dyneins to lose their connection with the neighbouring DMT. As a result, we do not observe density for the microtubule-binding domains of the axonemal dyneins, but we included them in our models for completeness. Homology models of motor domains were positioned by fitting into the observed cryo-EM densities for the linker, individual AAA+ modules and, where possible, the base of the stalk. Loss of connection to the neighbouring DMT may have also caused subtle changes in the conformation of the dynein motors relative to intact cilia. Small adjustments of the model may therefore be needed to model subtomogram averages of intact axonemes. In addition, although we observed two different conformational states for the *C. reinhardtii* β-HC, as described previously[5], we have modelled only the orientation in which the stalks extend towards the proximal end, because this conformation is most common in ODA structures from intact axonemes[74].

To construct the model of the axoneme's 9 + 2 architecture (Fig. 1b), we used subtomogram averages of DMT1 (EMD-2113), DMT2-8 (EMD-2132), DMT9 (EMD-2118)[3] and the central apparatus (EMD-31143)[75], and EMD-6756 (ref. 29) was used to guide the placement of DMTs relative to one another.

The map of the human 96-nm repeat was initially fitted with two copies of the pre-existing model of the human DMT 48-nm repeat (PDB: 7UNG)[64]. This revealed an additional MIP, RIBC1, which had been incorrectly assigned as RIBC2 in the 48-nm repeat. Building of the external axonemal complexes was then guided by the model of the *C. reinhardtii* 96-nm repeat and published models of the bovine ODA-DC (PDB: 7RRO)[37] and mouse RS head (PDB: 7DMP)[76]. Human homologues of *C. reinhardtii* axonemal proteins were identified from the results of BLAST searches (available from http://chlamyfp.org/)[77]. Subcomplexes of the human proteins were predicted by AlphaFold2 multimer

and placed in the cryo-EM density. A full list of positioned proteins is provided in Supplementary Table 3. The rationale for the placement of proteins without clear homology is given in Supplementary Table 4.

For both structures, fit-to-map was optimized using real-space refinement implemented in Coot, Namdinator[78] and Phenix[79]. The computational memory requirements needed to refine the 96-nm-long atomic model of the *C. reinhardtii* DMT meant that all-atom refinement could not be performed. Instead, three subregions were refined individually before being merged into a final PDB file. The smaller human model was refined as a single PDB file in Phenix. Atomic displacement factors for both models were refined using Phenix[79]. Side chains of asparagine, glutamine and histidine residues were automatedly flipped to improve hydrogen-bonding networks. Model statistics for the 96-nm modular repeat were generated using phenix.molprobity[80] and are provided in Extended Data Table 1.

## Figures

Figure panels and videos displaying cryo-EM maps or atomic models were generated using ChimeraX[63]. Maps coloured by local resolution were generated using RELION. Multiple-sequence alignments and phylogenetic trees were calculated using Clustal Omega[81]. Graphs were plotted in GraphPad Prism (GraphPad Software). Odds ratios (Fig. 3a) were calculated using R v4.0.3. Software used in the project was managed by SBGrid[82].

## Materials availability

Genomic samples and sequencing data obtained from patients are stored at University College London. Airway cells are biobanked in the UCL Living Airway Biobank administered through the UCL Great Ormond Street Institute of Child Health.

## Reporting summary

Further information on research design is available in the Nature Portfolio Reporting Summary linked to this article.

## Data availability

A composite cryo-EM map of the 96-nm repeat unit of DMTs from *C. reinhardtii* flagella has been deposited to the Electron Microscopy Data Bank (EMDB; https://www.ebi.ac.uk/pdbe/emdb/) with the accession code EMD-40220. The atomic model of the *C. reinhardtii* DMT has been deposited in the Protein Data Bank (PDB; https://www.rcsb.org/) with accession code 8GLV. A composite cryo-EM map of the 96-nm repeat unit of DMTs from human respiratory cilia has been deposited in the EMDB with accession code EMD-35888. The atomic model of the human DMT has been deposited in the PDB with accession code 8J07. For both EMD-40220 and EMD-35888, half maps of the individual axonemal complexes and the masks used for focused refinement have been deposited as additional maps associated with the entries. Source data are provided with this paper.

## Code availability

Data and code used to calculate the correlation between open and closed states of the ODA are available at https://doi.org/10.5281/zenodo.6908250.

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

**Acknowledgements** We thank C. Hogg and A. Rutman for TEM of nasal brushings, P. Griffin for acquiring high-speed video microscopy data, and R. K. Rai and P. Griffin for assistance in culturing human airway epithelial cells. Cryo-EM data were collected at the Harvard Center for Cryo-EM with assistance from S. Sterling, R. Walsh and M. Mayer. Cryo-EM data processing was supported by SBGrid (J. O'Connor, S. Rawson and W. Temple) and HMS Research Computing. T.W. was supported by a Helen Hay Whitney postdoctoral fellowship. M.G. was supported by a Charles A. King Trust postdoctoral fellowship. H.M.M. acknowledges funding from NIHR GOSH BRC, the Ministry of Higher Education in Egypt and MRC UCL Confidence in Concept (CiC7). A.B. was supported by grants from the National Institutes of Health (GM141109 and GM143183) and the Pew Charitable Trusts.

**Author contributions** T.W. performed all work on the *C. reinhardtii* axoneme. M.G. performed all structural work related to human axonemes. T.B. and S.V. prepared human axonemes from cultures. M.R.F. and H.M.M. performed genetic analysis. R.A.H., E.H. and C.O'C. provided clinical phenotype and cilia functional analysis. M.B. performed motility analysis. T.W. and A.B. wrote the paper. All authors approved the final content.

**Competing interests** The authors declare no competing interests.

**Additional information**
**Correspondence and requests for materials** should be addressed to Alan Brown.

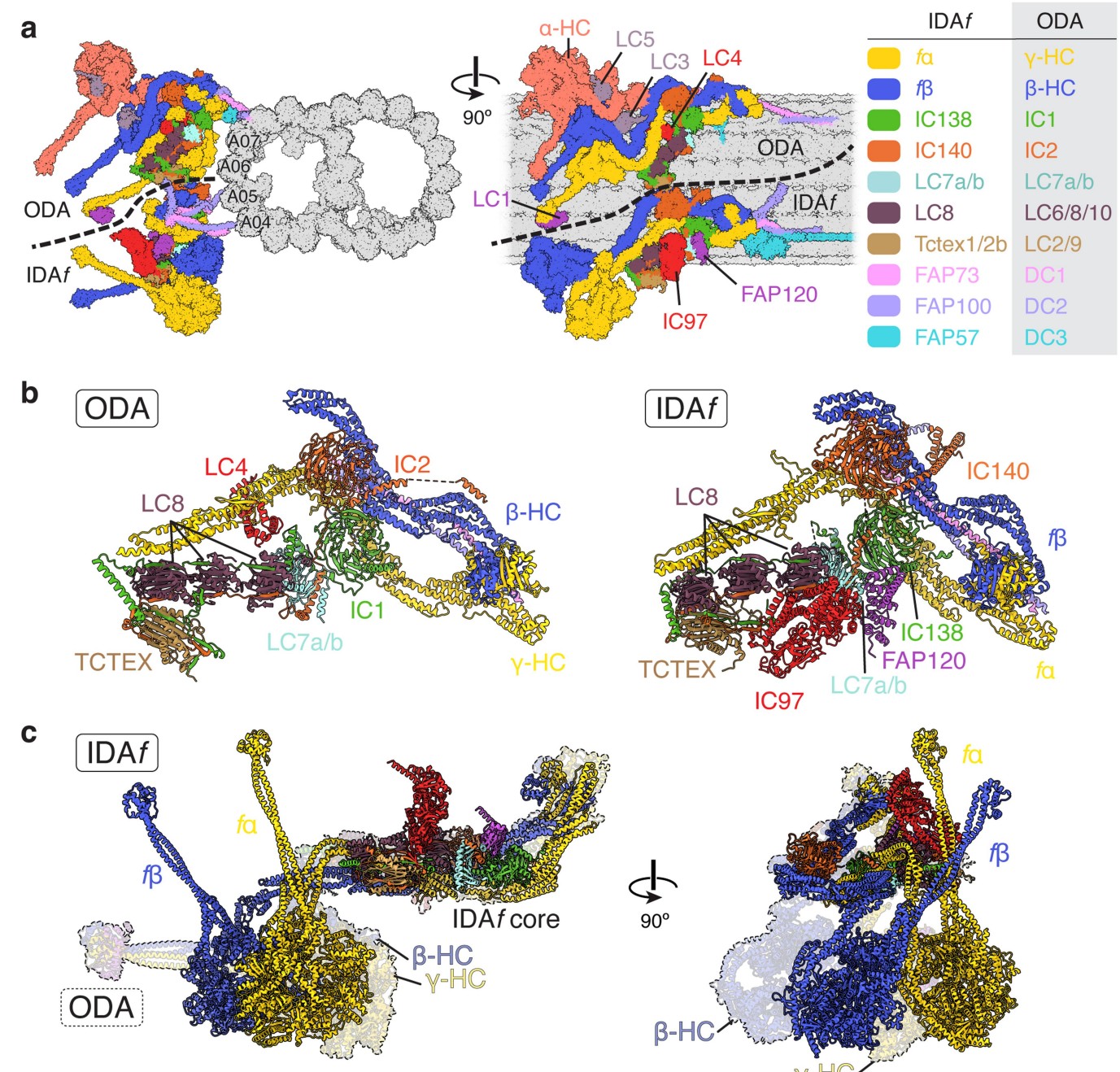

**Extended Data Fig. 1 | Structural comparison of inner dynein arm *f* (IDA*f*) and outer dynein arm (ODA) from *C. reinhardtii*. a**, Atomic model showing outer dynein arm 1 (ODA1) and inner dynein arm *f* (IDA*f*) bound to a doublet microtubule (DMT, gray). Dynein subunits are colored based upon their structural or functional similarity and listed in the table. Non-shared subunits are labeled. All other external complexes have been removed for clarity. **b**, Comparison of the cores of ODA (left) and IDA*f* (right). Paralogous subunits in these two structures occupy similar positions. IDA*f*-specific subunits IC97 and FAP120 bind the LC tower, whereas the ODA-specific LC4 binds γ-HC.

**c**, Comparison of motor domain orientation in IDA*f* and ODA. The ODA, faded and outlined by a dashed line, is overlaid by IDA*f*. The atomic models were aligned by superposing their cores. α-HC of ODA is omitted for clarity. Compared to ODA, the helical bundles of IDA*f* between the core region and the motors are less curled, resulting in rotation of the motors by 45–65° and displacement by 10 nm. These conformations differ from the active state of cytoplasmic dynein[83], suggesting that the connection between the core and the motor may help specialize each dynein for distinct roles.

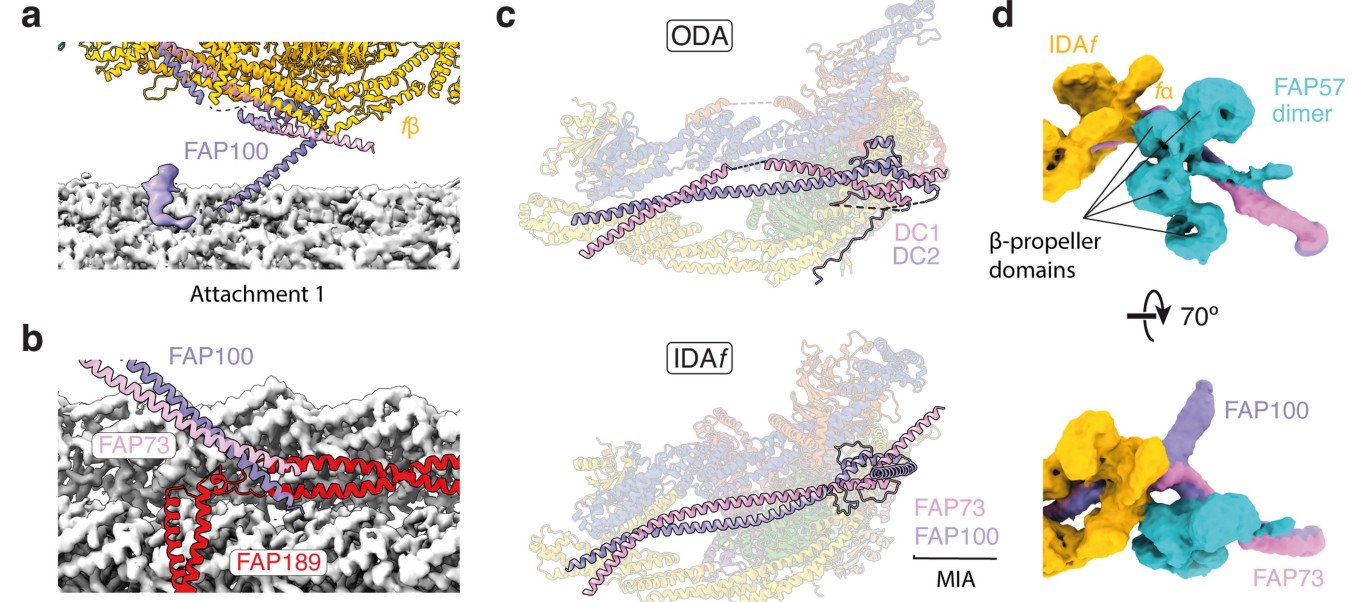

**Extended Data Fig. 2 | The *C. reinhardtii* MIA complex contributes to the docking of inner dynein arm *f* (IDA*f*) to the doublet microtubule (DMT).** **a**, MIA (a heterodimer of FAP73/FAP100) interacts with protofilament A05 of the DMT through a C-terminal helix of FAP100 beneath the *f*β heavy chain tail. Unmodeled density of FAP100 is also shown. **b**, The N-terminus of the MIA coiled coil interacts with a FAP189 family member near the site of its 90° bend on the DMT surface. **c**, Comparison of the binding of docking complex subunits DC1/2 to ODA and MIA to IDA*f*. Both coiled coils bisect the helical bundles of the dynein heavy chains within the dynein cores. **d**, Cryo-EM map showing the interaction between the *f*α tail, MIA, and the β-propeller domains of a FAP57 homodimer.

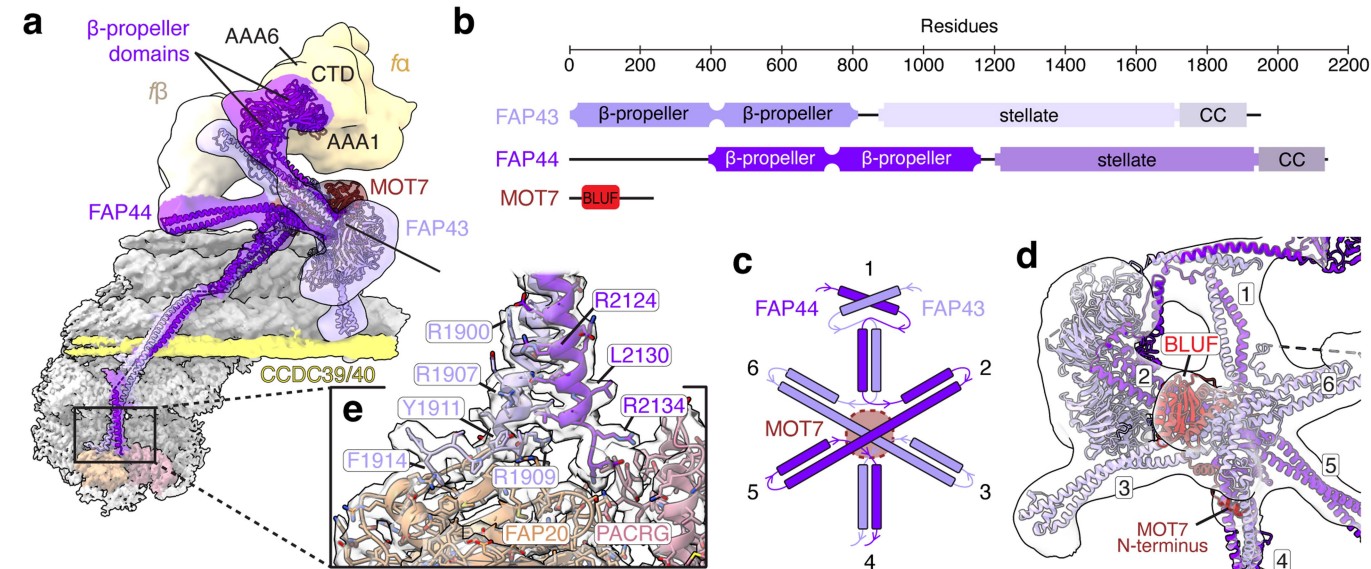

**Extended Data Fig. 3 | Structure of the *C. reinhardtii* tether/tetherhead (T/TH) complex. a**, Composite cryo-EM map colored by T/TH subunit and the motor domains of inner dynein arm (IDA) *f*α and *f*β. The doublet microtubule surface is shown in gray. The N-terminal pair of β-propellers of FAP44 interact with the AAA1, AAA6, and C-terminal domains of the *f*α motor. **b**, Domain organization of T/TH subunits FAP43, FAP44, and MOT7. Both FAP43 and FAP44 contain an N-terminal pair of β-propellers, followed by a helical stellate structure, and a microtubule-associated coiled coil (CC). MOT7 contains a

BLUF domain that is proposed to sense blue light[12]. **c**, Schematic illustrating the six-armed stellate structure of T/TH with MOT7 at the center. **d**, Reverse side of the stellate showing the position of MOT7 and its BLUF domain (bright red) within the cryo-EM map. The resolution is too low to observe any flavin cofactors. On the left, the N-terminal pair of β-propellers from FAP43 bind arm 2 of the stellate. **e**, Interaction of the FAP43/44 coiled coil with the inner junction proteins FAP20 and PACRG. Bulky sidechain residues used for identification are labeled.

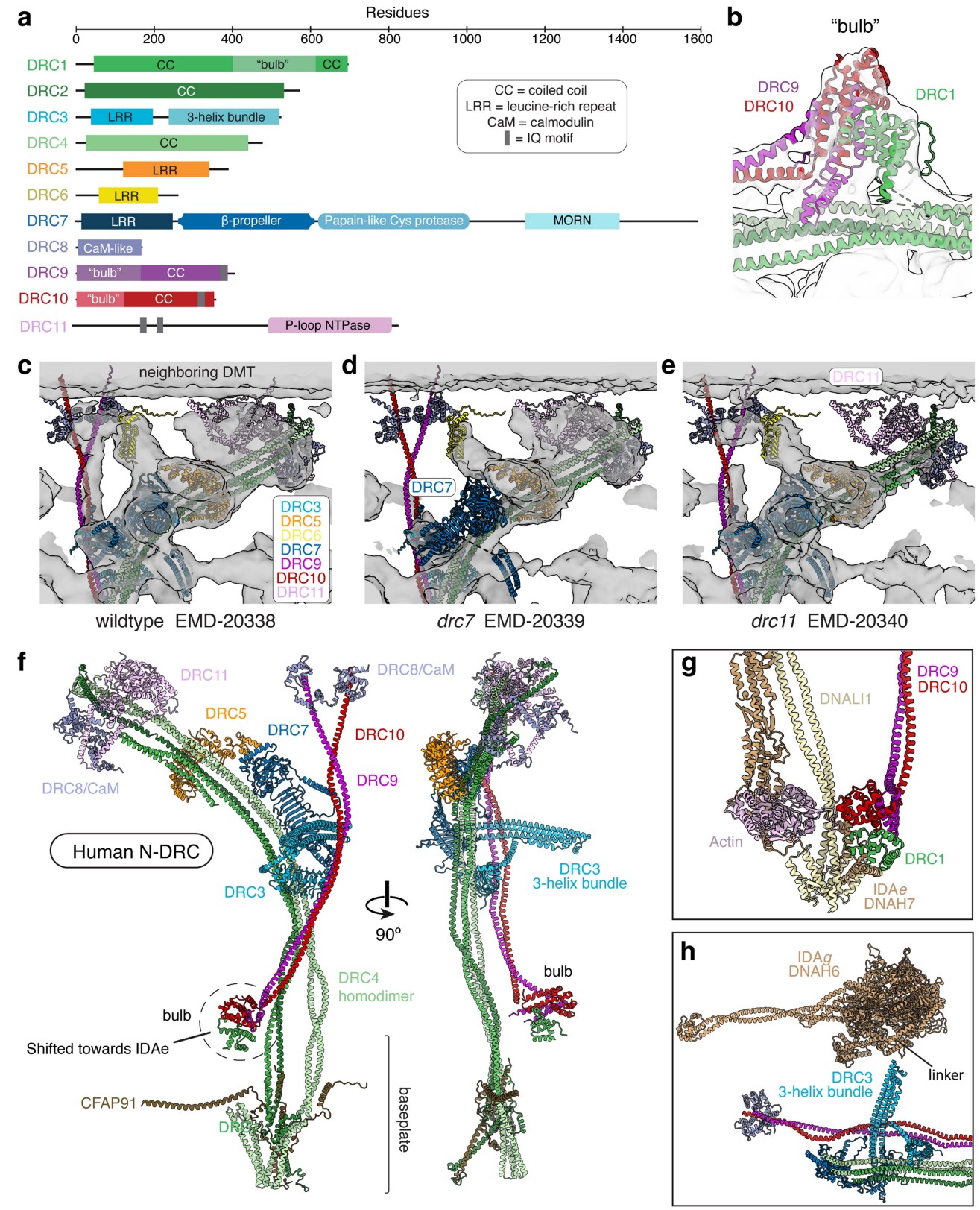

**Extended Data Fig. 4** | See next page for caption.

**Extended Data Fig. 4 | Integrative modeling of the nexin-dynein regulatory complex (N-DRC) from *C. reinhardtii* and *H. sapiens*. a**, Domain architecture of *C. reinhardtii* N-DRC subunits. **b**, Architecture of the N-DRC "bulb" in *C. reinhardtii*. **c**, Model of the N-DRC linker in the cryo-ET subtomogram average of the wildtype *C. reinhartii* axoneme (EMD-20338)[27]. Small differences between our model and the cryo-ET map are likely due to the loss of interactions with the neighboring doublet microtubule (DMT) in our splayed axoneme cryo-EM sample. **d**, Model of the N-DRC linker in the subtomogram average of the *drc7* mutant axoneme (EMD-20339)[27]. The DRC7 subunit resides outside of the cryo-ET density, consistent with the loss of DRC7 in the mutant axonemes. Reduced density of the distal lobe, containing the C-termini of DRC9 and 10, suggest an altered or weakened interaction between DRC9/10 and the neighboring DMT in this mutant. **e**, Model of the N-DRC linker in the subtomogram average of the *drc11* mutant axoneme (EMD-20340)[27]. Reduced density of the proximal lobe is consistent with the position of DRC11. **f**, Orthogonal views of the atomic model of human N-DRC. The bulb of human N-DRC (DRC1/9/10) is uncoupled from the baseplate unlike in *C. reinhardtii*. **g**, Interaction between the bulb and the N-terminal tail domain of DNAH7 (IDA*e*). **h**, Interaction between the 3-helix bundle of DRC3 and the linker of DNAH6 (IDA*g*).

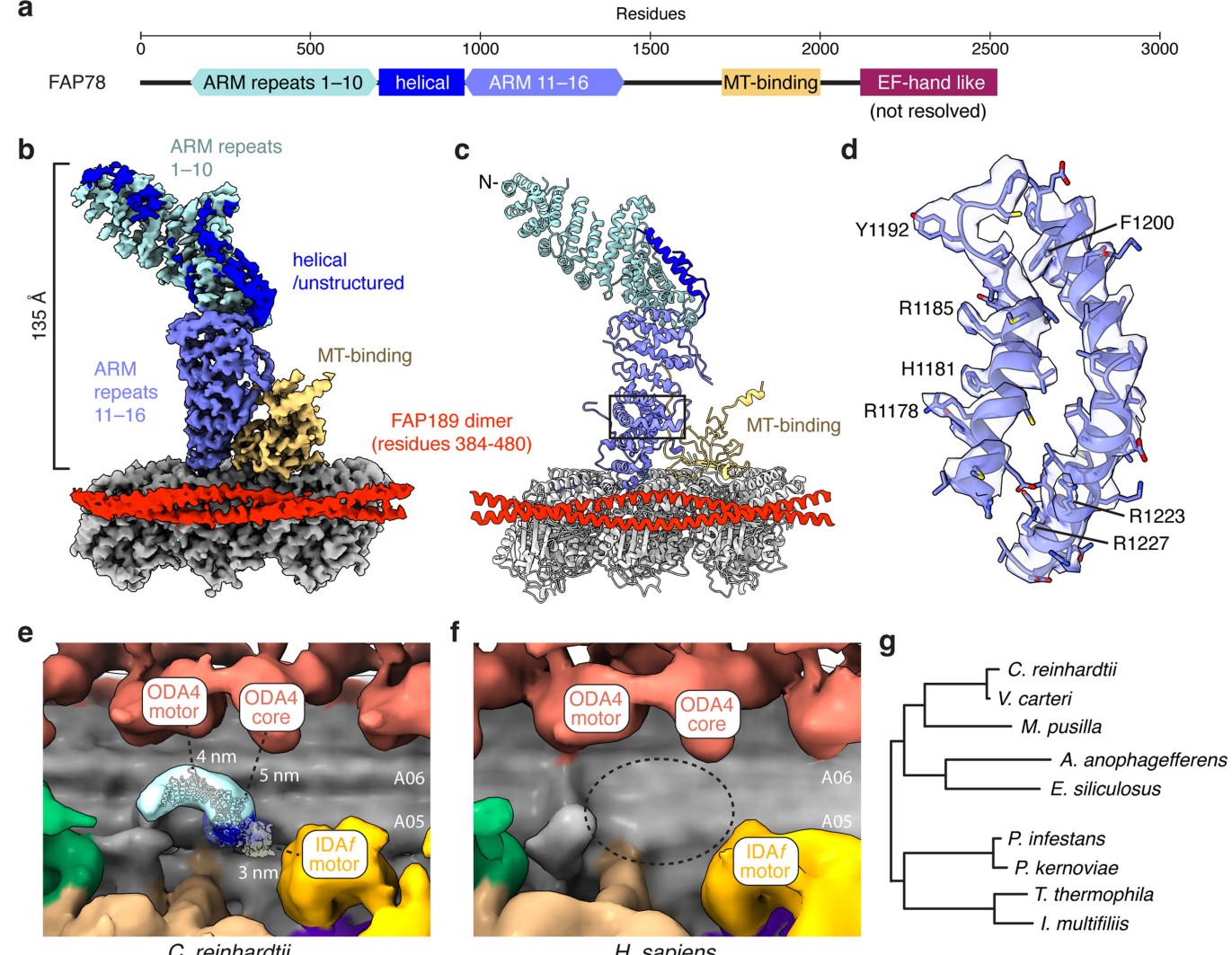

**Extended Data Fig. 5 | Identification of FAP78 as the distal protrusion (DP) of the *C. reinhardtii* axoneme. a**, Domain architecture of FAP78. FAP78 contains 16 armadillo (ARM) repeats interrupted with a helical/unstructured region. The C-terminal half contains a microtubule (MT) binding domain and an EF-hand like domain, which is not resolved in our cryo-EM map. **b**, Cryo-EM map of the DP colored by domain. **c**, Atomic model of FAP78. **d**, Sidechain density of a section of FAP78 boxed in panel c. **e**, The FAP78 model positioned within the subtomogram average of the *C. reinhardtii* axoneme (EMD-6872)[24]. The height of the DP is 155 Å in this map. FAP78 is located within 3–5 nm of IDA*f* and ODA4,

suggesting a possible role in dynein regulation, although we do not observe direct interactions. **f**, The DP is absent from axonemes from human respiratory cilia (EMD-5950)[32]. The expected location is denoted by a dashed oval. The subtomogram averages in panels e and f are colored according to the color scheme established in Fig. 1. **g**, Phylogenetic distribution of FAP78 is limited to green algae and other single-celled ciliates. The homolog in *T. thermophila* only aligns to the C-terminal half of FAP78 and does not contain ARM repeats, which likely explains why a distal protrusion is not observed in subtomogram averages of *T. thermophila* axonemes (EMD-12119)[84].

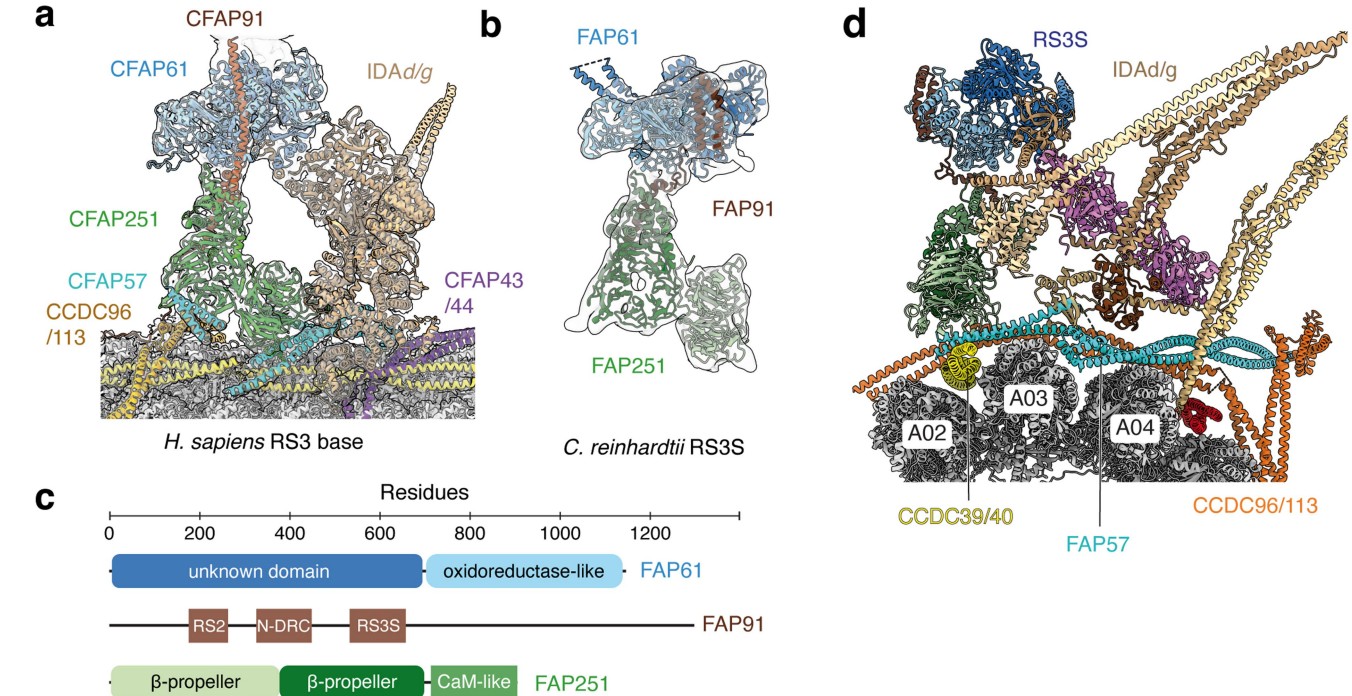

**Extended Data Fig. 6 | Radial spoke 3 (RS3). a**, Model and cryo-EM map showing the base of human RS3 and its association with IDA*d/g*. Coiled coils from CCDC96/113 and CFAP57 are docking factors for RS3. CFAP43/44 (components of the T/TH complex) are docking factors for IDA*d/g*. **b**, Model and cryo-EM map of *C. reinhardtii* RS3S. **c**, Domain organization of RS3S subunits. The boxes on FAP91 indicate consecutive binding sites for RS2, the N-DRC and RS3S. Both the oxidoreductase-like domain of FAP61 and the calmodulin (CaM)-like domain of FAP251 interact with FAP91. **d**, Atomic model of the docking factors for RS3S in *C. reinhardtii*.

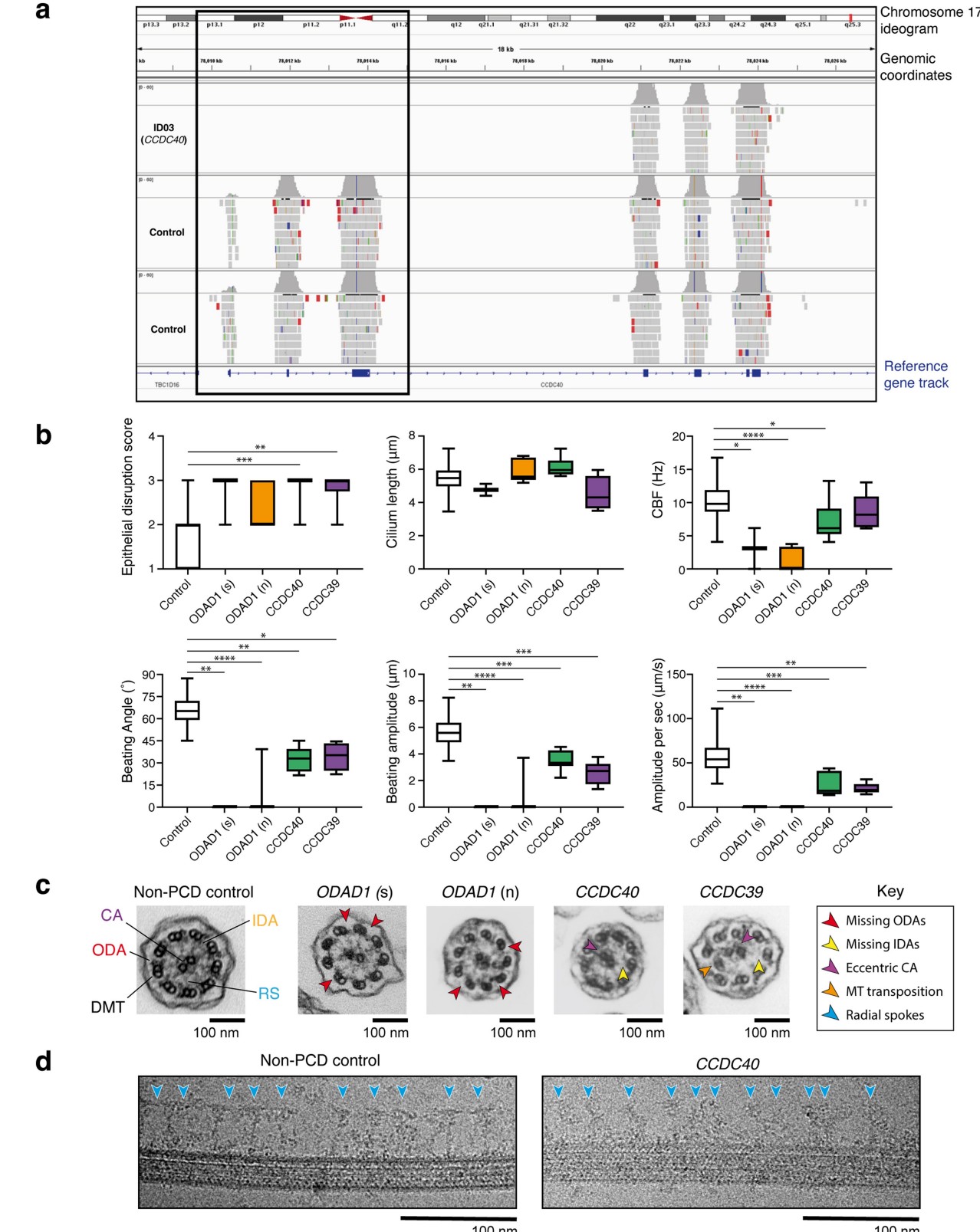

**a**

Chromosome 17 ideogram

Genomic coordinates

ID03 (*CCDC40*)

Control

Control

Reference gene track

**b**

**c**

Non-PCD control    *ODAD1* (s)    *ODAD1* (n)    *CCDC40*    *CCDC39*    Key

▶ Missing ODAs
▶ Missing IDAs
▶ Eccentric CA
▶ MT transposition
▶ Radial spokes

100 nm    100 nm    100 nm    100 nm    100 nm

**d**

Non-PCD control    *CCDC40*

100 nm    100 nm

**Extended Data Fig. 7** | See next page for caption.

**Extended Data Fig. 7 | Primary ciliary dyskinesia (PCD) patient information.**
**a**, Integrative Genomics Viewer (IGV) snapshot illustrating the homozygous exon 1–3 deletion in the *CCDC40* gene of patient ID03. **b**, Analysis of high-speed video data from PCD patients showing defects in epithelia consistency, cilia beat frequency, beat angle, and amplitude compared to healthy, non-PCD controls. The box represents the interquartile range, the center represents the median, and the whiskers represent minimum and maximum measurements. Statistical significance was determined by one-way ANOVA (non-parametric Kruskal-Wallis), with * $P \leq 0.05$, ** $P \leq 0.01$, *** $P \leq 0.001$ and **** $P \leq 0.0001$. The number of movies used for the analysis were 104 (control); 3 (*ODAD1* s); 8 (*ODAD1* n); 8 (*CCDC40*), and 6 (*CCDC39*). For each mutation, the movies came from the same patient. *ODAD1* (s) refers to individual ID01 with a splice mutation, and *ODAD1* (n) refers to individual ID02 with a nonsense mutation in *ODAD1*. **c**, Representative clinical transmission electron microscopy (TEM) images selected from a minimum of >100 screened cross-sections showing axoneme cross sections from a non-PCD control (with axonemal complexes labeled), two individuals with mutations in *ODAD1* (ID01 and ID02), an individual with a mutation in *CCDC40* (ID03), and an individual with a mutation in *CCDC39* (ID04). Red arrows mark the absence of outer dynein arms (ODAs). Orange arrows mark microtubule (MT) transposition. Yellow arrows mark the absence of inner dynein arms (IDAs). Purple arrows mark an abnormal central apparatus (CA). **d**, Compared to a non-PCD control, radial spokes (blue arrows) attached to doublet microtubules in individual ID03 (with a *CCDC40* mutation) are intermittently bound, consistent with the loss of strict 96-nm periodicity. These micrographs sections were selected from a total of 37,071 non-PCD control micrographs, and 3,483 *CCDC40* micrographs.

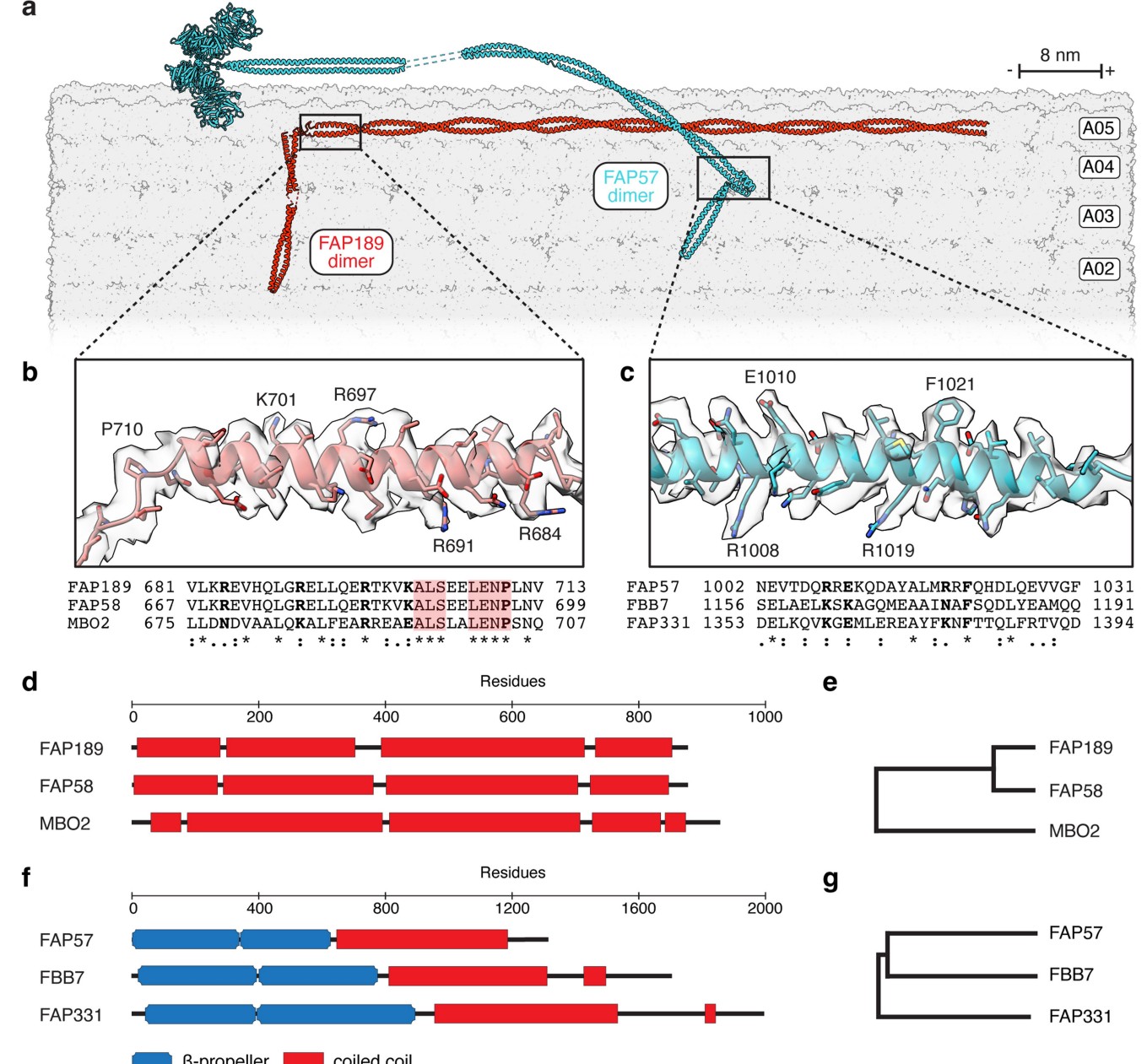

**a**

8 nm

FAP57 dimer

FAP189 dimer

A05
A04
A03
A02

**b**

P710
K701
R697
R684
R691

```
FAP189  681  VLKREVHQLGRELLQERTKVKALSEEELENPLNV  713
FAP58   667  VLKREVHQLGRELLQERTKVKALSEEELENPLNV  699
MBO2    675  LLDNDVAALQKALFEARREAEALSLALENPSNQ  707
             :*..:*    *  :  *::  *  *.:***  ****  *
```

**c**

E1010
F1021
R1008
R1019

```
FAP57   1002  NEVTDQRREKQDAYALMRRFQHDLQEVVGF  1031
FBB7    1156  SELAELKSKAGQMEAAINAFSQDLYEAMQQ  1191
FAP331  1353  DELKQVKGEMLEREAYFKNFTTQLFRTVQD  1394
              .*:  :  :  :    :    *  :.  *    :*  ..:
```

**d**

Residues

FAP189

FAP58

MBO2

**e**

FAP189
FAP58
MBO2

**f**

Residues

FAP57

FBB7

FAP331

**g**

FAP57
FBB7
FAP331

β-propeller    coiled coil

**Extended Data Fig. 8 | Coiled-coil docking factors potentially contribute to axonemal asymmetry. a**, Atomic models of the FAP189 and FAP57 homodimers. Other axonemal complexes are omitted for clarity. FAP189 forms an L-shaped coiled coil that occupies the interprotofilament cleft between protofilaments A04 and A05. **b**, Fit of FAP189 atomic model into the cryo-EM map. Labeled amino acids are denoted below by bold font in the multiple sequence alignment of FAP189 with its paralogs FAP58 and MBO2. The conserved sequence ALSXXLENP, highlighted in red, corresponds to residues that interact with FAP100 of the MIA complex (see Extended Data Fig. 2b). **c**, Agreement of cryo-EM sidechain density with the atomic model of FAP57. Residues labeled in the model are denoted by bold font in the multiple sequence alignment below. **d**, Domain architecture of FAP189, FAP58, and MBO2. **e**, Phylogenetic tree showing the relationship between FAP189, FAP58, and MBO2. FAP189 and FAP58 share 80% sequence identity. **f**, Domain architecture of FAP57, FBB7 and FAP331, which are composed of tandem N-terminal β-propellers and a long, helical C-terminal region. **g**, Phylogenetic tree showing the relationship between FAP57, FBB7, and FAP331.

**Extended Data Table 1 | Cryo-EM data collection, refinement and validation statistics**

| | 96-nm repeat of the *C. reinhardtii* flagella axoneme (EMD-40220) (PDB 8GLV) | 96-nm repeat of the *H. sapiens* respiratory cilia axoneme (EMD-35888) (PDB 8J07) |
|---|---|---|
| **Data collection and processing** | | |
| Facility | Harvard Cryo-EM Center for Structural Biology | Harvard Cryo-EM Center for Structural Biology |
| Microscope | Titan Krios 2 (Thermo Fisher Scientific) | Titan Krios 1 (Thermo Fisher Scientific) |
| Energy filter | BioQuantum K3 Imaging Filter with a slit width of 25 eV (Gatan) | BioQuantum K3 Imaging Filter with a slit width of 25 eV (Gatan) |
| Detector | K3 (Gatan) | K3 (Gatan) |
| Magnification | 64,000× | 64,000× |
| Voltage (kV) | 300 | 300 |
| Electron exposure (e–/Å$^2$) | 61-62 | 60 |
| Exposure time (s) | 3.7–3.8 | 3.76 |
| Defocus range (μm) | -0.5 to -2 | -0.8 to -2 |
| Pixel size (Å) | 1.36 | 1.37 |
| Symmetry imposed | C1 | C1 |
| Movie stacks (and after quality control) | 36,918 (31,275) | 37,071 (16,933) |
| Initial 8-nm particle images (no.) (and after quality control) | 8,546,747 (8,439,890) | 2,582,833 (2,582,833) |
| Final particle images (no.) | Variable | Variable |
| Map resolution range (Å) | 3.2 to 16.7 | 3.3 to 13.6 |
| FSC threshold | 0.143 | 0.143 |
| | | |
| **Refinement*** | | |
| Initial model used (PDB code) | PDBs: 6U42, 7JTS, 7JTK, 7JU4, 7KZM, 7KZO, 6L4P | PDBs: 8GLV, 7UNG, 7DMP |
| Resolution limit set in refinement (Å) | 5 | 5 |
| Map sharpening *B* factor (Å$^2$) | Variable | Variable |
| Correlation coefficient (CCmask) | 0.66 | 0.57 |
| Model composition | | |
| Chains | 1,177 | 976 |
| Non-hydrogen atoms | 3,968,237 | 3,163,619 |
| Protein residues | 503,117 | 396,235 |
| Ligands | 1,009 | 859 |
| *B* factors (Å$^2$) | | |
| Protein | 149.0 | 162.1 |
| Ligand | 78.2 | 58.2 |
| R.m.s. deviations | | |
| Bond lengths (Å) | 0.004 | 0.005 |
| Bond angles (°) | 1.02 | 1.14 |
| Validation | | |
| MolProbity score | 1.92 | 1.67 |
| Clashscore | 14.1 | 5.91 |
| Poor rotamers (%) | 0.12 | 0.03 |
| Ramachandran plot | | |
| Favored (%) | 96.09 | 96.59 |
| Allowed (%) | 3.88 | 3.35 |
| Disallowed (%) | 0.03 | 0.06 |

*Refinement and validation statistics for the 96-nm repeat of the *C. reinhardtii* flagella axoneme are the average (mean) of values generated after splitting the model into three regions and running each separately through Phenix.real_space_refine. Attempts to calculate statistics for the intact complex failed because of insufficient memory. The human 96-nm repeat is smaller and was refined as a single model.

**Extended Data Table 2 | Genetics and diagnostic results for all four PCD cases**

| | ID01 | ID02 | ID03 | ID04 |
|---|---|---|---|---|
| **Sex** | M | M | M | M |
| **Age** | 17 | 22 | 40 | 9 |
| **Gene** | *ODAD1* | *ODAD1* | *CCDC40* | *CCDC39* |
| **GRCh38** | chr19:g. 48303953:C:T | chr19:g.48312029:G:A | N/A | chr3:g.180631499:CTCTTCTC:CTC |
| **Transcript** | NM_001364171.2 | NM_001364171.2 | NM_017950.4 | NM_181426.2 |
| **cDNA** | c.853G>A | c.448C>T | exon 1-3 deletion | c.1964_1968delCTCTT |
| **Protein** | p.Ala285Thr | p.Arg150* | N/A | p.Glu655Glyfs*23 |
| **dbSNP** | rs147718607 | rs752269093 | N/A | N/A |
| **ClinVar** | VCV000039637.10 | VCV001324024.5 | N/A | VCV001070026.4 |
| **nNO (ppb)** | 10.5 | 7.7 | 7.0 | n.d. (too young to measure) |
| **TEM** | Missing ODAs | Missing ODAs | Lack of IDAs; 16% cross sections show disorganized and damaged MTs | Lack of IDAs; 21% cross sections show disorganized and damaged MTs |
| **Clinical** | Recurrent chest infections, continuous wet productive cough, persistent rhinorrhoea, *situs inversus totalis*, transposition large vessels. | Recurrent chest infections, continuous wet productive cough, wheeze, persistent rhinorrhoea, *situs inversus*. | Recurrent chest infections, continuous wet productive cough, intermittent rhinorrhoea, bronchiectasis, azoospermia. | Recurrent chest infections, continuous wet productive cough, wheeze, persistent rhinorrhoea. |
| **Citation** | Onoufriadis et al[42] | Davis et al[53] | This paper | This paper |

|---|---|

# Reporting Summary

## Statistics

For all statistical analyses, confirm that the following items are present in the figure legend, table legend, main text, or Methods section.

| n/a | Confirmed | |
|---|---|---|
| ☐ | ☒ | The exact sample size (*n*) for each experimental group/condition, given as a discrete number and unit of measurement |
| ☐ | ☒ | A statement on whether measurements were taken from distinct samples or whether the same sample was measured repeatedly |
| ☐ | ☒ | The statistical test(s) used AND whether they are one- or two-sided *Only common tests should be described solely by name; describe more complex techniques in the Methods section.* |
| ☒ | ☐ | A description of all covariates tested |
| ☒ | ☐ | A description of any assumptions or corrections, such as tests of normality and adjustment for multiple comparisons |
| ☐ | ☒ | A full description of the statistical parameters including central tendency (e.g. means) or other basic estimates (e.g. regression coefficient) AND variation (e.g. standard deviation) or associated estimates of uncertainty (e.g. confidence intervals) |
| ☒ | ☐ | For null hypothesis testing, the test statistic (e.g. *F*, *t*, *r*) with confidence intervals, effect sizes, degrees of freedom and *P* value noted *Give P values as exact values whenever suitable.* |
| ☒ | ☐ | For Bayesian analysis, information on the choice of priors and Markov chain Monte Carlo settings |
| ☒ | ☐ | For hierarchical and complex designs, identification of the appropriate level for tests and full reporting of outcomes |
| ☒ | ☐ | Estimates of effect sizes (e.g. Cohen's *d*, Pearson's *r*), indicating how they were calculated |

*Our web collection on statistics for biologists contains articles on many of the points above.*

## Software and code

Policy information about availability of computer code

| Data collection | SerialEM v3.6 and v3.7 |
|---|---|
| Data analysis | RELION-3.1;  RELION-4.0; Chimera v1.15; ChimeraX v1.3; Coot v0.9.4.1; phenix.real_space_refine v1.19.2-4158; Phenix.molprobity v1.19.2-4158; SWISS-MODEL; AlphaFold v2.1.1; AlphaFold v2.2.0; R v4.0.3; MotionCor2; CTFFIND4; Namdinator (no version number); PDBeFOLD v2.59; Clustal Omega Web Service (no version number); DeepTracer Web Service (no version number); Matlab R2022a; GraphPad Prism v. 8.0.0, Zenodo (https://doi.org/10.5281/zenodo.6908250). |

For manuscripts utilizing custom algorithms or software that are central to the research but not yet described in published literature, software must be made available to editors and reviewers. We strongly encourage code deposition in a community repository (e.g. GitHub). See the Nature Portfolio guidelines for submitting code & software for further information.

## Data

Policy information about availability of data

All manuscripts must include a data availability statement. This statement should provide the following information, where applicable:
- Accession codes, unique identifiers, or web links for publicly available datasets
- A description of any restrictions on data availability
- For clinical datasets or third party data, please ensure that the statement adheres to our policy

A composite cryo-EM map of the 96-nm repeat unit of DMTs from C. reinhardtii flagella has been deposited to the Electron Microscopy Data Bank (EMDB; https://

## Human research participants

Policy information about studies involving human research participants and Sex and Gender in Research.

| | |
|---|---|
| Reporting on sex and gender | Sex or gender was not considered in the study design. All human cilia samples were obtained from males. |
| Population characteristics | Nasal brushings were collected from four individuals with PCD. These individuals were aged 9, 17, 22, and 40. All individuals were determined to have PCD using standard diagnostic tests. All had recurrent chest infections, continuous wet productive coughs and persistent rhinorrhoea. |
| Recruitment | The samples with genotypes of interest were obtained retrospectively from patients recruited as part of ongoing genetic research projects. |
| Ethics oversight | Peripheral blood samples for genetic screening, and respiratory epithelial cells collected from the inferior nasal turbinate by nasal scrape biopsy, were obtained from affected individuals and healthy volunteers under ethical approval granted through the Health Research Authority London Bloomsbury Research Ethics Committee (REC 08/H0713/82; IRAS 103488) and Living Airway Biobank, administered through the UCL Great Ormond Street Institute of Child Health (REC 19/NW/0171, IRAS 261511, Health Research Authority North West Liverpool East Research Ethics Committee). Informed written consent was obtained from all participants prior to enrollment in the study. |

Note that full information on the approval of the study protocol must also be provided in the manuscript.

# Field-specific reporting

Please select the one below that is the best fit for your research. If you are not sure, read the appropriate sections before making your selection.

☒ Life sciences ☐ Behavioural & social sciences ☐ Ecological, evolutionary & environmental sciences

For a reference copy of the document with all sections, see nature.com/documents/nr-reporting-summary-flat.pdf

# Life sciences study design

All studies must disclose on these points even when the disclosure is negative.

| | |
|---|---|
| Sample size | 1. For cilia beat analysis, movie recordings from 11 non-PCD controls and 4 PCD patients were used. Number of movies = 104 (control); 3 (ODAD1 splice); 8 (ODAD1 nonsense); 8 (CCDC40), and 6 (CCDC39). No methods were used to predetermine sample size.<br>2. For cryo-EM processing, no methods were used to predetermine sample size. Final structures of the C. reinhardtii and H. sapiens 96-nm repeat were calculated from a total of 31,275 and 16,933 micrographs, respectively. The final number of micrographs was determined to be sufficient as they provided structures of doublet microtubules to better than 4 Angstrom resolution.<br>3. Sample size for odds ratio calculations were determined by unsupervised 3D classification using RELION software to generate 160,444 particles for ODA1, 162,179 particles for ODA2, 214,203 particles for ODA3, and 177,762 particles for ODA4. |
| Data exclusions | 1. For the cilia beat analysis, static cilia were excluded from the analysis.<br>2. The algorithms used for single particle image processing may down-weight or exclude particles as part of their refinement strategy. |
| Replication | 1. Multiple cryo-EM datasets were collected for the wild-type doublet microtubule structures. Their reconstructions were consistent, and merged.<br>2. For the ciliary beat analysis, between 3 and 104 different movies were analyzed (see Sample Size above). |
| Randomization | For calculation of the Fourier Shell Correlation (FSC), cryo-EM particles were randomly split into two halves using the RELION software. |
| Blinding | 1. The initial diagnostic assessment was blinded, as these tests were done before a confirmed genetic result. As up to 80% of patients referred for diagnostic testing are found not to have PCD, there should not be any bias to influence these results.<br>2. Blinding was not used for the cilia beat analysis. However, the methods are objective and should not introduce bias or impact the measurements.<br>3. Blinding was not used for the cryo-EM studies; the researchers were aware of the identity of the samples. However, similar processing steps were used for the wild-type and PCD samples. |

# Reporting for specific materials, systems and methods

We require information from authors about some types of materials, experimental systems and methods used in many studies. Here, indicate whether each material, system or method listed is relevant to your study. If you are not sure if a list item applies to your research, read the appropriate section before selecting a response.

## Materials & experimental systems

| n/a | Involved in the study |
|-----|----------------------|
| ☒ ☐ | Antibodies |
| ☒ ☐ | Eukaryotic cell lines |
| ☒ ☐ | Palaeontology and archaeology |
| ☒ ☐ | Animals and other organisms |
| ☒ ☐ | Clinical data |
| ☒ ☐ | Dual use research of concern |

## Methods

| n/a | Involved in the study |
|-----|----------------------|
| ☒ ☐ | ChIP-seq |
| ☒ ☐ | Flow cytometry |
| ☒ ☐ | MRI-based neuroimaging |

