## [Peer Review File · Nature]

Manuscript Title: Axonemal structures reveal mechanoregulatory and disease mechanisms.

Reviewer Comments & Author Rebuttals

Reviewer Reports on the Initial Version:

Referees' comments:

Referee #1 (Remarks to the Author):

Motile cilia, a beating organelle in eukaryotes, have a long history of research. It evokes biophysical interest, how it beats and how the waveform (snake-like symmetric form in sperm, arch-shaped asymmetric in respiratory tract and conical rotational on embryo) is determined. A number of question at cellular biological level are open, such as origin of nine-fold symmetrical spoke-like structure ("9+2"), why the cytoplasmic part (basal body) has triplet microtubules and the extracellular part (axoneme) has doublet microtubules. It also links to disease – defect of motile cilia causes lung disease, brain disorder, infertility in male and female and developmental problems, called primary ciliary dyskinesia (PCD). Since 1950's, motile cilia have been investigated intensively in the various aspects, biochemistry, genetics, biophysics, structural biology, cell biology, to answer these questions. In the recent years, a number of proteomic and structural works appeared, step-by-step revealing components of cilia (>400 proteins known in the axoneme; Pazour et al. 2005 J. Cell Biol. 170, 103) and their 3D arrangement, but it was still limited to ~50 component proteins. Entire view of this organelle has been awaited.

This seminal work, Walton et al. from the Brown group, solved almost all the component proteins in the axonemal doublet microtubule, employing single particle cryo-EM. They treated 96nm periodic unit of the doublet microtubule as one single particle and applied atomic resolution cryo-EM technique. In the past by the same approach, the Brown group solved the structure of microtubule luminal inner proteins (MIPs) (Ma et al. 2019 Cell 179, 909), BBSome (Singh et al. 2020 Elife 9:e53322), the radial spoke (Gui et al. 2021 NSMB 28, 29), the outer dynein arm (Walton et al. 2021 12, 477) and the central singlet microtubule (Gui et al. 2022 NSMB 29, 483), leading the field of cilia structural biology. Nevertheless this study culminates their achievement, covering >100 component proteins from green algae *Chlamydomonas* (most popular model organism in cilia research) and nearly hundred components from human cilia. If you count multiple copies (such as tubulin) involved in one structure, structure of several hundred proteins are included in each structure. We could not imagine atomic resolution structural analysis of a complex with so many components and high flexibility. They made full use of image analysis technique, as well as AI-based model building to achieve this work. Their approach and process are sound and reliable, while this reviewer would request further clarification of technical points (later).

This manuscript provides all the structural basis, which future works will refer to answer molecular mechanism of ciliary function. In this sense, like the human genome project, history of motile cilia research will be divided into "pre- Walton et al." and "post- Walton et al.". While various biological stories (such as "odds model" in Fig.4) mentioned in this manuscript are speculative hypotheses and rather for limited interest of experts working on cilia structure, this work will be cited by those who design mutants, conduct molecular dynamics, cellular imaging or physiological analysis. With this reason, this reviewer is confident that this manuscript should be published in Nature.

Presentation and graphics are very well organized and impressive. Experimental procedure is clearly stated.

They also investigated structure of DMT from PCD patients. This part is rather premature (detail below) and this reviewer would recommend the authors to publish this part, including the section "External microtubule periodicities are independent", Fig.7, Extended Data Fig.7 and 8, separately elsewhere, believing that this work is worth for publication in Nature without that part.

Here this reviewer suggests a few points to improve this manuscript to present their phenomenal structure in more informative ways.

Fig.5, 6: Do the authors consider conformation of dyneins physiological? If they are in the same conformation as in the axoneme, their microtubule binding domains (MTBD) should be located to bind to the protofilament, between alpha and beta tubulins (Redwine et al. 2012 Science 337, 1532). Does this structure fit to this geometry.

Outer dynein arm: In their past work (Walton et al. 2021), the stalk of the beta dynein was oriented toward the distal direction, while the stalks of the alpha and gamma dyneins extend toward the proximal direction. However, in the present work (Fig.1A) all the stalks extend toward the proximal end. Is this correct? If so, what caused this change?

Another question about ODA: Are the two-head ODA from human same as *Chlamydomonas* ODA minus alpha-HC?

Differences between *Chlamydomonas* and human cilia in structure must be listed in detail. Many small structures show differences. For example, in Fig.6ab, p38, p44 and TTC29, ZMYND12 look different. A supplementary table of corresponding components, as well as about individual proteins, such as one protein from human is shorter than the corresponding *Chlamydomonas* protein, will help.

Dynein f: IC138 and IC97 were hypothesized to regulate dynein f by phosphorylation (Hendrickson et al. 2004 MBC 15, 5431; Bower et al. 2009 MBC 20, 3055; Wirschell et al. 2009 MBC 20, 3044). Do the authors have any thought about this?

Inner dyneins: Microtubule scientists will be interested to see detailed views of inner dynein tails' being anchored on the A-tubule. Are the tail-B-DMT interface same among various inner dyneins? Are there any other protein binding to the microtubule in the same folding?

Actin at the base of many inner dyneins, whereas in the past mutant work (Kamiya 2002 Int. Rev. Cytol. 219, 115), only dynein a, c, d, e had a defect in the absence of actin gene. Is there any thought why dynein b and g stay on DMT without actin? Is the conformation of actin same as monomeric or filamentous actin?

External protein: Is there any other IDA/CCDC39+40 interface than DNAH12? Does *Chlamydomonas* have any protein corresponding to it?

N-DRC: Their findings of K-rich DRC head structure is intriguing. Is there any thought how DRC binds to the B-tubule? How is surface charge on B-tubule?

Extended Data Fig.2de: Here the authors provided two groups of reconstructions "MT proximal regions" and "MT distal regions". But it is not mentioned in the text. If they reconstructed 3D structure of the 96nm units from the proximal and the distal regions separately, it could be extended further as interesting findings and discussion, as different ODA components are known between these areas (Dougherty et al. 2016 Am J. Respir. Cell Mol. Biol. 55, 213). But for that they should provide corresponding structures between d and e. Or are they for other purposes (comparison of resolution, for example)?

As mentioned above, this reviewer recommends publication of their results from PCD patients elsewhere. Their conclusion that 24nm and 96nm periodicities are independent of each other was already proved (at the same level as in this work) by *Chlamydomonas* mutants. ODA deficit mutants, such as *oda1* (Takada et al. 2002 MBC 13, 1015; Ishikawa et al. 2007 J. Mol. Biol. 368, 1249), keeps 96nm periodicity. Genetically engineered mutants in which 96nm periodicity was disturbed have ODA with 24nm spacing (Oda et al. 2014 Science 346, 857).

If they still pursue publishing their PCD structure, the data must be processed correctly. Forcing 8nm or 48nm to the structure which actually has 96nm periodicity will cause artifact. Ideally cryo-ET/96nm subtomogram averaging is the right way to reconstruct structure, when there are not enough number of cilia. But probably it is difficult to prepare cryo-sample from those patients again. This reviewer wonders if their micrographs can be reanalyzed with 96nm periodicity.

Maheshwari et al. (2015) Structure 23, 1584 did single particle analysis of doublets, starting from fewer micrographs than this work, using outdated instrument, to obtain structures of most of MIPs. Hopefully the authors can reconstruct similar 96-nm unit structure by similar hand-picking approach, which is more reliable to conclude how the MIPs are maintained in the PCD patients.

They also have to clarify the link between their structure and clinical data (Extended Data Fig. 7),

for example by showing the place of mutation and genetic defect on the atomic structure.

Citation: The reference list is reasonable, but often each reference is cited in a wrong place. It is probably systematic errors during editing. The authors should check throughout the manuscript. Here are examples (but likely not limited to them).

p.4, line8: ref3-5 are not papers on dynein f motility. They meant 13-15. But ref13-15 are papers showing deletion mutant of dynein f are still motile, not showing dynein f does not generate power against B-MT.

p.4, line25: Kutomi et al. did not study FAP57. Maybe they meant 33.

p.4, 4 from bottom: MOT7 paper is 30, not 33.

MIA is also cited wrongly. (p.6, line2)

Extended data fig.6: ref21 is not structure paper.

p.8, line8: ref16, 47 are not IDA papers.

Other suggestions regarding references.

DRC: Oda et al. (2015) Mol. Biol. Cell 26, 294 provided the best structural knowledge of N-DRC so far, by BCCP tagging and cryo-ET. Their N-DRC model should be examined after atomic-resolution model by this work.

p.7, line12: It was known for a long time that human has ODA with 2 heavy chains and three RSs, much before from Nicastro paper.

RS head paper by the Vale group (Grossman-Haham et al. 2021 NSMB 28, 20) should be referred in the context of human RS head structure.

Other minor points:

p.8 line1 "human is not light sensitive": Kutomi et al. 2022 studied *Chlamydomonas* MOT7 and its homologue from *Ciona*, which is not light sensitive. So the function of MOT7 might not be limited to phototaxis.

p.8, line11: not clear what they did by "bioinformatics deduction".

Referee #2 (Remarks to the Author):

A stunning piece of cryoET defining nanoscale architecture of the green algae motile axoneme, through to the respiratory cilia of human airway cells in healthy controls and those of patients with the rare genetic motile ciliopathy primary ciliary dyskinesia. It is a very weighty paper with breath-taking detail and resolution, which really captures how this 'parts list' of components necessary for cilia motility are coordinated in time and space, and importantly across species, to allow for an effective stroke necessary for propulsion or generation of fluid flow. In humans, these processes are fundamental at all stages of life- from how we reproduce, to early embryonic patterning, to maintaining healthy airways and nervous systems- cilia motility is at the heart of it all. This paper with revisions is highly suitable to publication in Nature- it will be 'THE' landmark reference paper for those in biophysics, genetics and cell biology interested in the machinery of cilia motility.

As a cell biologist and developmental geneticist, I cannot comment in detail on the technical aspects of the cryoET work, but make some suggestions on accessibility and more pointed questions on the human disease work.

Major points

1) P3 You make the statement your 'cryoEM-based approach can reveal the disease mechanisms'- this is a stretch, you had selected genetically defined and fairly well mechanistically understood null PCD alleles a posteriori. Your EM constructions were not done agnostically to define which components were missing when a particular gene was mutated- they were not blinded. Without genetics, it remains unknown whether your approach would be sufficient to diagnose or reveal disease mechanisms. It is possible, but not demonstrated here.

2) The text jumps between *Chlamy* and human data. Is Extended Data Fig.1 entirely *Chlamy*- if so state? b. jumps to human- it is very confusing. Extended Fig. 3 same- likely *Chlamy* as a-HC is not shown.

- 3) Why not show that the 24 nm spacing of ODAs is unaffected in the CCDC39/40 mutants axoneme averages- you say in the text that these are 'unaffected' why not show it graphically along a longitudinal view- is the periodicity even subtly altered without the 96nm ruler (Figure 7c)? This work is something that could actually resolve a lot of this. Are both of these 'rulers' in fact completely independent?
- 4) You make a grand statement in the discussion about 'help de-orphanize dynein HCs in other species' but what about within the same species? Can you detect the 'compartmentalization' proximal-distally of your ODA heavy chains in the human respiratory cilia- the change from DNAH11 to DNAH9? Supplementary Data Table 4 suggests you have only DNAH9- discuss? Do other such regional differences in MIPS or RSs etc?

Minor points

- 1) For clarity, when first describing the IDAs (end of page 2) maybe explicitly state that there are 6 single headed dyneins and a single double headed one which each appear within the 96 nm repeat once. For non-specialists, this is dense and tricky going.
- 2) For a general audience, it may be worthwhile stating what the levels of resolution are within the field prior to this study (end of page 3). What resolution is necessary for the 'most complete atomic model of the axoneme to date'?
- 3) P6, 'a 155-Å tall tower' is too colloquial and confusing for this dataset.
- 4) P8, human respiratory cilia there are 6 isoforms of human actin. Does your modelling tell you which one it is for IDAa and IDAc?
- 5) P9, when you speak of coding mutations like the 'c.853G>A substitution in ODAD1' I believe the ODAD1 needs to be italicized, likely for all the genes in this paragraph.
- 6) P9, the text reads awkwardly 'by patient genome sequencing analysis' by 'analysis of patient genome sequencing'.
- 7) Docking complex mutants should not affect cilia number- why do the authors state 'too few cilia could be obtained for imaging'. In contrast, CC3C39/40 mutants have as their table says lots of microtubule disorganization and one would have thought would be more difficult to image- this is not the case, they are higher resolution. Can the authors explain why?
- 8) In your discussion, you highlight 'striking species-specific differences'- you undersell the fact your methodology could be applied to look at cilia-specific differences within the same species, say airway cells and ependymal cilia lining the brain, highlighting the possible molecular foundations for differences to sensitivity to dysfunction when specific genes are mutated.
- 9) Figure 7a- how many samples, what is plotted? Is this an average? Statistics on the humans should be referenced within figure legends.
- 10) Extended Data Fig.2a- why are the numbers of particles lower (and lower resolution for IDAb) given the repeats should all be fairly similar in how they are averaged?
- 11) Extended Data Fig 4. Love the London tube map but explain why some lines end without a 'station' complex- i.e. IDAb on 2 RS3S on 6?
- 12) What does 'Epithelial disruption score' mean in Extended Data Fig.7c Y axis label?
- 13) Extended Data Fig. 8: CCDC40 patient ID125 and ODAD1 ID25 patient sections are too dark to resolve any of the ultrastructure- need to show features being highlighted.

Edits

- 1) Typo Methods: Clinical transmission electron microscopy- should be 'Sorenson's'
- 2) Extended Data Fig.7 legend 'PCD CCDC40 mutation patient' awkward- consider 'PCD patient with CCDC40 mutation'?

Referee #3 (Remarks to the Author):

The manuscript reports the first nearly complete structural model of the 96-nm repeat of the axoneme from *Chlamydomonas reinhardtii* flagella with all the major structural components included. The whole model includes the pseudo-atomic structures of the inner dynein arms (IDAs)

and nexin-dynein regulatory complexes (N-DRCs) that are newly determined in the manuscript by single particle cryo-EM, in addition to previously determined structures of outer dynein arms (ODAs), the microtubule tubule doublet complex, the radial spoke complex and the central pair complex that were published by the authors' lab and others. The structural map calculations were nicely done – they utilized single particle cryo-EM with local masks to reconstruct the axonemal component complexes. Some of the structural model building referenced the protein structures predicted by AlphaFold. The new structural information provides important insight into the dynein docking and beating regulatory mechanisms that involve DRC and the newly identified outer-inner dynein links.

Moreover, the manuscript determined the pseudoatomic models of axonemal 96nm repeats for human respiratory cilia from four patients of primary ciliary dyskinesia (PCD), which could offer direct structural interpretation on how the PCD mutations result in disease phenotypes. The structural comparison of the human and *Chlamydomonas* axonemes also revealed some species-dependent structural differences, particularly in IDAs. These comparisons provide new data and insights for understanding motile cilia evolution. Overall, the manuscript presents excellent structural biology work with a large amount of research effort involved. The whole structural models of the 96 nm repeats of motile cilia will serve as structure baselines for further mechanistic understanding of motile cilia beating and regulation. In addition, the paper itself is well-organized and clearly written. I recommend it for publication. I have no major concern except for the following minor suggestion to the authors.

This is a comprehensive structural report for a complicated molecular machine with many proteins and protein complexes. For the human axonemes, as summarized in Extended Data Fig 2a, different structural maps are shown with different resolutions ranging from 3.3 to 13.4Å. It would be helpful if another column is added behind the resolution column to give a brief description of the methods used for model building with each of the maps. Such a table should also be provided for the axoneme of *Chlamydomonas* flagellar structural maps and model building because it would help the readers develop an accuracy expectation for each part of the whole structural model.

Referee #1 (Remarks to the Author):

Motile cilia, a beating organelle in eukaryotes, have a long history of research. It evokes biophysical interest, how it beats and how the waveform (snake-like symmetric form in sperm, arch-shaped asymmetric in respiratory tract and conical rotational on embryo) is determined. A number of questions at cellular biological level are open, such as origin of nine-fold symmetrical spoke-like structure (“9+2”), why the cytoplasmic part (basal body) has triplet microtubules and the extracellular part (axoneme) has doublet microtubules. It also links to disease – defect of motile cilia causes lung disease, brain disorder, infertility in male and female and developmental problems, called primary ciliary dyskinesia (PCD). Since 1950’s, motile cilia have been investigated intensively in the various aspects, biochemistry, genetics, biophysics, structural biology, cell biology, to answer these questions. In the recent years, a number of proteomic and structural works appeared, step-by-step revealing components of cilia (>400 proteins known in the axoneme; Pazour et al. 2005 J. Cell Biol. 170, 103) and their 3D arrangement, but it was still limited to ~50 component proteins. Entire view of this organelle has been awaited.

This seminal work, Walton et al. from the Brown group, solved almost all the component proteins in the axonemal doublet microtubule, employing single particle cryo-EM. They treated 96nm periodic unit of the doublet microtubule as one single particle and applied atomic resolution cryo-EM technique. In the past by the same approach, the Brown group solved the structure of microtubule luminal inner proteins (MIPs) (Ma et al. 2019 Cell 179, 909), BBSome (Singh et al. 2020 Elife 9:e53322), the radial spoke (Gui et al. 2021 NSMB 28, 29), the outer dynein arm (Walton et al. 2021 12, 477) and the central singlet microtubule (Gui et al. 2022 NSMB 29, 483), leading the field of cilia structural biology. Nevertheless this study culminates their achievement, covering >100 component proteins from green algae *Chlamydomonas* (most popular model organism in cilia research) and nearly hundred components from human cilia. If you count multiple copies (such as tubulin) involved in one structure, structure of several hundred proteins are included in each structure. We could not imagine atomic resolution structural analysis of a complex with so many components and high flexibility. They made full use of image analysis technique, as well as AI-based model building to achieve this work. Their approach and process are sound and reliable, while this reviewer would request further clarification of technical points (later). This manuscript provides all the structural basis, which future works will refer to answer molecular mechanism of ciliary function. In this sense, like the human genome project, history of motile cilia research will be divided into “pre- Walton et al.” and “post- Walton et al.”. While various biological stories (such as “odds model” in Fig.4) mentioned in this manuscript are speculative hypotheses and rather for limited interest of experts working on cilia structure, this work will be cited by those who design mutants, conduct molecular dynamics, cellular imaging or physiological analysis. With this reason, this reviewer is confident that this manuscript should be published in Nature. Presentation and graphics are very well organized and impressive. Experimental procedure is clearly stated.

They also investigated structure of DMT from PCD patients. This part is rather premature (detail below) and this reviewer would recommend the authors to publish this part, including the section “External microtubule periodicities are independent”, Fig.7, Extended Data Fig.7 and 8, separately elsewhere, believing that this work is worth for publication in Nature without that part.

Here this reviewer suggests a few points to improve this manuscript to present their phenomenal structure in more informative ways.

Fig.5, 6: Do the authors consider conformation of dyneins physiological? If they are in the same conformation as in the axoneme, their microtubule binding domains (MTBD) should be located to bind

to the protofilament, between alpha and beta tubulins (Redwine et al. 2012 Science 337, 1532). Does this structure fit to this geometry.

During the splaying of axonemes into separate DMTs, the majority of axonemal dyneins lose their connection with the neighboring DMT. We therefore do not observe density for the MTBDs but have included them in the models for completeness. Loss of connection to the neighboring DMT may have also caused subtle changes to the conformation of the dynein motors compared to in intact cilia. Although our models are based on our cryo-EM data, they can be used to model subtomogram averages of intact axoneme through approaches recently used to examine ODA conformation (Zimmermann *et al.*, bioRxiv, 10.1101/2022.05.02.490280). We have added this information to the Methods section.

Outer dynein arm: In their past work (Walton et al. 2021), the stalk of the beta dynein was oriented toward the distal direction, while the stalks of the alpha and gamma dyneins extend toward the proximal direction. However, in the present work (Fig.1A) all the stalks extend toward the proximal end. Is this correct? If so, what caused this change?

We still observe both conformations but have only modeled the orientation in which the stalks extend toward the proximal end because this conformation appears most common in ODA structures from intact axonemes (Zimmermann *et al.*, 2022; 10.1101/2022.05.02.490280). We have added this information to the Methods section.

Another question about ODA: Are the two-head ODA from human same as *Chlamydomonas* ODA minus alpha-HC?

The overall architecture and arrangement of the ODA core subdomains from the two species are highly similar, except that *Chlamydomonas* has an additional LC4 subunit that has no equivalent in humans. The docking complexes are also different, but this difference was described previously in the report on the bovine ODA-DC (Gui *et al.*, 2021; 10.1016/j.cell.2021.10.007), which closely resembles the human ODA-DC. Because of the differences in the docking complexes, the ODAs in *Chlamydomonas* and human have slightly different orientations with respect to the DMT surface.

Differences between *Chlamydomonas* and human cilia in structure must be listed in detail. Many small structures show differences. For example, in Fig.6ab, p38, p44 and TTC29, ZMYND12 look different. A supplementary table of corresponding components, as well as about individual proteins, such as one protein from human is shorter than the corresponding *Chlamydomonas* protein, will help.

We have listed all *Chlamydomonas* external proteins in Supplementary Table 1 and all human external proteins in Supplementary Table 4. These tables include the protein length and molecular masses for each protein, allowing readers to compare these properties between orthologs. (MIPs are compared in Gui *et al.*, 2022; 10.1073/pnas.2207605119). Every protein shows small differences between the two species – for example *Chlamydomonas* p38/p44 and human TTC29/ZMYND12 adopt slightly different conformations but are superposable as single proteins – but listing all these differences would be too much information and detract from our more significant findings. Instead, we have focused on describing the largest differences between the two structures, such as the N-DRC and IDA composition.

Dynein f: IC138 and IC97 were hypothesized to regulate dynein f by phosphorylation (Hendrickson et al. 2004 MBC 15, 5431; Bower et al. 2009 MBC 20, 3055; Wirschell et al. 2009 MBC 20, 3044). Do the authors have any thought about this?

Previous studies have suggested that IC97 is required for the effect of IC138 hyperphosphorylation on inhibition of microtubule sliding, but have not identified the specific sites of IC138 phosphorylation, which complicates structural interpretation. Possibly, IC138 phosphorylation might regulate the interaction between IDA_f with IDA_α and the neighboring DMT, since IC97 and the N-terminus of IC138 are located in this region by our structures.

Inner dyneins: Microtubule scientists will be interested to see detailed views of inner dynein tails' being anchored on the A-tubule. Are the tail-B-DMT interface same among various inner dyneins? Are there any other protein binding to the microtubule in the same folding?

When IDAs interact directly with the A tubule, it is almost exclusively through the p28/DNALI1 subunit, which recognizes the tubulin interdimer interface. This interaction is very similar between IDAs within a species (e.g., human IDA_α, *b*, *c*, *e*) and between species (e.g., comparing IDA_c in *Chlamydomonas* and human). We were unable to find another example in the literature of a similar fold interacting with tubulin, although the interdimer interface is a common binding site for microtubule-associated proteins. The only other possible direct interaction between an IDA and tubulin is between Arg44 of human DNAH12 (IDA_α) and the α -tubulin C-terminal tail, but this is unlikely to make a significant contribution to docking.

Actin at the base of many inner dyneins, whereas in the past mutant work (Kamiya 2002 Int. Rev. Cytol. 219, 115), only dynein a, c, d, e had a defect in the absence of actin gene. Is there any thought why dynein b and g stay on DMT without actin?

In *ida5* mutants, actin is replaced by an actin-related protein (NAP1) that is 64% identical to actin (Lee *et al.*, 1997; 10.1016/s0378-1119(97)00254-0 and Kato-Minoura *et al.*, 1998; 10.1006/bbrc.1998.9373). It is possible that, because IDA heavy chains interact differently with actin, NAP1 can compensate for actin in IDA_b and *g* but not the other IDAs. Further work, outside the scope of this study, will be needed to test this hypothesis.

Is the conformation of actin same as monomeric or filamentous actin?

As shown in the figure to the left, the conformation of actin associated with IDAs more closely matches G-actin (PDB: 1ATN) than F-actin (PDB: 8A2S).

External protein: Is there any other IDA/CCDC39+40 interface than DNAH12? Does *Chlamydomonas* have any protein corresponding to it?

DNAH12 is the only human IDA heavy chain that interacts with the CCDC39/40 molecular ruler (as shown in Fig. 6c). We have not identified any direct connections between IDAs and CCDC39/40 in *Chlamydomonas*.

N-DRC: Their findings of K-rich DRC head structure is intriguing. Is there any thought how DRC binds to the B-tubule? How is surface charge on B-tubule?

We believe that the interaction between the K-rich DRC head structure and the B-tubule is mediated entirely through electrostatic interactions with the negatively charged polyglutamylate groups. We have edited the text to make this clearer.

Extended Data Fig.2de: Here the authors provided two groups of reconstructions “MT proximal regions” and “MT distal regions”. But it is not mentioned in the text. If they reconstructed 3D structure of the 96nm units from the proximal and the distal regions separately, it could be extended further as interesting findings and discussion, as different ODA components are known between these areas (Dougherty et al. 2016 Am J. Respir. Cell Mol. Biol. 55, 213). But for that they should provide corresponding structures between d and e. Or are they for other purposes (comparison of resolution, for example)?

We apologize for any confusion these terms introduced. We are not referring to distal and proximal regions of the axoneme, but regions proximal and distal to the microtubule surface. For example, a “MT proximal region” would be the radial spoke base, whereas a “MT distal region” would be the radial spoke head. We have replaced “proximal” with “MT-associated” and “distal” with peripheral, to avoid confusion. We are unable to distinguish proximal and distal regions of the cilium, such as differences in the ODA heavy chains.

As mentioned above, this reviewer recommends publication of their results from PCD patients elsewhere. Their conclusion that 24nm and 96nm periodicities are independent of each other was already proved (at the same level as in this work) by *Chlamydomonas* mutants. ODA deficit mutants, such as *oda1* (Takada et al. 2002 MBC 13, 1015; Ishikawa et al. 2007 J. Mol. Biol. 368, 1249), keeps 96nm periodicity. Genetically engineered mutants in which 96nm periodicity was disturbed have ODA with 24nm spacing (Oda et al. 2014 Science 346, 857).

We appreciate the reviewer's suggestion to publish our PCD patient results elsewhere. However, we and editor believe that this part of the study adds an extra dimension to understanding the independence between periodicities in human axonemes and helping augment clinical diagnostics in PCD. Although prior studies using *C. reinhardtii* have shown a similar phenomenon of independent periodicities they were less comprehensive. For example, Oda et al. (2014) did not report whether the 48-nm MIP organization was disrupted in response to disrupted 96-nm periodicity. Additionally, there are significant differences between the 24, 48 and 96-nm periodicities in human and *Chlamydomonas* axonemes, so it is not necessarily expected that the results would be the same. We have revised the text to better distinguish our results from prior studies, highlighting that ours are studies with bona fide PCD-causing mutations in humans rather than random or synthetically engineered mutants in a model organism. We have also added an additional reference to Oda et al (2014) when comparing our data with prior data from *Chlamydomonas*.

Moreover, our work has medical implications that the prior studies with *Chlamydomonas* did not. By demonstrating that cryo-EM can be used to visualize structural changes that arise from human PCD

mutants, our work takes human axonemal imaging into the translational dimension by introducing a new diagnostic method with the potential to observe defects missed by conventional transmission electron microscopy screens.

If they still pursue publishing their PCD structure, the data must be processed correctly. Forcing 8nm or 48nm to the structure which actually has 96nm periodicity will cause artifact.

Although we agree that there are limitations to our approach, we do believe that our reconstructions still provide valuable insights into the affected structures including identifying missing complexes and revealing whether complete periodicities are lost. For the *CCDC39* and *CCDC40* mutants, 48-nm periodicity is the highest periodicity that can be reconstructed because 96-nm periodicity is lost – it is only the *ODAD1* mutants that are reconstructed to a lower periodicity than what is potentially possible. To ensure the accuracy of our results and avoid artifacts, we have reconstructed DMTs from wild-type cilia using the same periodicities, similar numbers of particles, and resolved to similar resolutions. This approach allowed us to make a direct comparison with the affected structures. We have also made sure to clarify this important control in the text.

Ideally cryo-ET/96nm subtomogram averaging is the right way to reconstruct structure, when there are not enough number of cilia. But probably it is difficult to prepare cryo-sample from those patients again.

Cryo-ET followed by subtomogram averaging would be feasible but negate the purpose of the study which was to show that cryo-EM with single-particle analysis can be used to reconstruct structures of DMTs from patient cilia. Prior work has already successfully established that cryo-ET and subtomogram averaging can be used (*Lin et al.*, 2014; 10.1038/ncomms6727 and *Zhao et al.*, 2021; 10.1091/mbc.E20-12-0806). Furthermore, as the reviewer suggests, collecting tomography data for each of the four mutants wouldn't be possible without recontacting the patients.

This reviewer wonders if their micrographs can be reanalyzed with 96nm periodicity. Maheshwari et al. (2015) *Structure* 23, 1584 did single particle analysis of doublets, starting from fewer micrographs than this work, using outdated instrument, to obtain structures of most of MIPs. Hopefully the authors can reconstruct similar 96-nm unit structure by similar hand-picking approach, which is more reliable to conclude how the MIPs are maintained in the PCD patients.

Although the method of Maheshwari et al. (2015) is an interesting approach, it would not work for the *CCDC39* and *CCDC40* mutants, as these are missing the 96-nm periodicity of the radial spokes necessary to hand-pick particles. Moreover, picking particles based on radial spokes risks introducing orientation bias into the 3D reconstruction that could lead to artifacts in the density map. The advantage of our approach is that it starts by aligning particles using the intrinsic 8-nm periodicity of tubulin without requiring any prior assumptions about periodicity or needing to be able to visualize any of the external components which can be difficult to distinguish in raw micrographs.

They also have to clarify the link between their structure and clinical data (Extended Data Fig. 7), for example by showing the place of mutation and genetic defect on the atomic structure.

We thank the reviewer for the suggestion and have added a panel to Fig. 7 (panel c) highlighting the location of *CCDC39*, *CCDC40* and *ODAD1* in the atomic model of the human DMT. The exact locus of each mutation is reported in Extended Data Fig. 8a. In each case, the predicted effect of the mutation is a complete loss of protein due to degradation (as proven in the case of *ODAD1* c.853G>A; Onoufriadis et

al., 2013; 10.1016/j.ajhg.2012.11.002) rather than expression of a mutant protein. This is due to either protein frameshift with premature termination (in the case of *ODAD1* p.Ala248Thrfs52*, the *ODAD1* large 3-exon deletion, or the *CCDC39* 5-basepair deletion p.Glu655Glyfs*23), or due to predicted nonsense-mediated decay associated with premature stop codons (in the case of the *ODAD1* p.Arg150* mutation). To make this clear, we have added a sentence to the text.

Citation: The reference list is reasonable, but often each reference is cited in a wrong place. It is probably systematic errors during editing. The authors should check throughout the manuscript. Here are examples (but likely not limited to them).

p.4, line8: ref3-5 are not papers on dynein f motility. They meant 13-15. But ref13-15 are papers showing deletion mutant of dynein f are still motile, not showing dynein f does not generate power against B-MT.

p.4, line25: Kutomi et al. did not study FAP57. Maybe they meant 33.

p.4, 4 from bottom: MOT7 paper is 30, not 33.

MIA is also cited wrongly. (p.6, line2)

Extended data fig.6: ref21 is not structure paper.

p.8, line8: ref16, 47 are not IDA papers.

The reference discrepancies appear to have been caused by the reference list in the Supplemental Table not matching the reference list in the main text. We have now made sure the lists are consistent. We have also checked the paper to make sure every reference is correct and appropriate.

Other suggestions regarding references.

DRC: Oda et al. (2015) Mol. Biol. Cell 26, 294 provided the best structural knowledge of N-DRC so far, by BCCP tagging and cryo-ET. Their N-DRC model should be examined after atomic-resolution model by this work.

The work of Oda *et al* (2015) successfully placed the N-termini of DRC1, 2 and 4 near the neighboring DMT and the C-termini at the baseplate. They also identified DRC5 in the central region. We have added this reference to the main text.

p.7, line12: It was known for a long time that human has ODA with 2 heavy chains and three RSs, much before from Nicastro paper.

We apologize if there is a paper that we have overlooked, but we could not find an earlier one that conclusively demonstrates that human respiratory cilia have 3 radial spokes per 96-nm repeat and a two-headed ODA. We have edited the text, so that we are no longer implying the Lin *et al.*, 2014; 10.1038/ncomms6727 paper was the first.

RS head paper by the Vale group (Grossman-Haham et al. 2021 NSMB 28, 20) should be referred in the context of human RS head structure.

We apologize for this omission and have added the reference to the paper.

Other minor points:

p.8 line1 “human is not light sensitive”: Kutomi et al. 2022 studied *Chlamydomonas* MOT7 and its homologue from *Ciona*, which is not light sensitive. So the function of MOT7 might not be limited to phototaxis.

We thank the reviewers for this important point. We have corrected our phrasing.

p.8, line11: not clear what they did by “bioinformatics deduction”.

We are referring to the bioinformatics analysis reported by Kollmar, 2016; 10.1093/molbev/msw213 in which they identified the heavy chains of human IDAf (DNAH2 and DNAH10), IDAd (DNAH1) and IDAg (DNAH6). We have edited the text to improve the clarity of this paragraph.

Referee #2 (Remarks to the Author):

A stunning piece of cryoET defining nanoscale architecture of the green algae motile axoneme, through to the respiratory cilia of human airway cells in healthy controls and those of patients with the rare genetic motile ciliopathy primary ciliary dyskinesia. It is a very weighty paper with breath-taking detail and resolution, which really captures how this ‘parts list’ of components necessary for cilia motility are coordinated in time and space, and importantly across species, to allow for an effective stroke necessary for propulsion or generation of fluid flow. In humans, these processes are fundamental at all stages of life- from how we reproduce, to early embryonic patterning, to maintaining healthy airways and nervous systems- cilia motility is at the heart of it all. This paper with revisions is highly suitable to publication in Nature- it will be 'THE' landmark reference paper for those in biophysics, genetics and cell biology interested in the machinery of cilia motility.

As a cell biologist and developmental geneticist, I cannot comment in detail on the technical aspects of the cryoET work, but make some suggestions on accessibility and more pointed questions on the human disease work.

Major points

1) P3 You make the statement your ‘cryoEM-based approach can reveal the disease mechanisms’- this is a stretch, you had selected genetically defined and fairly well mechanistically understood null PCD alleles a posteriori. Your EM constructions were not done agnostically to define which components were missing when a particular gene was mutated- they were not blinded. Without genetics, it remains unknown whether your approach would be sufficient to diagnose or reveal disease mechanisms. It is possible, but not demonstrated here.

We appreciate this comment and have reduced the claim by modifying the statement to “we demonstrate how cryo-EM of axonemal microtubules derived from patient cells can add structural detail to help elucidate the disease mechanisms that cause ciliopathies”. We did not mean to imply that cryo-EM can be used for diagnosis and the identification of disease mechanisms without genetics or other diagnostic information. Instead, we see cryo-EM as a tool that can provide additional information to what is currently available. For example, genetics can identify where in a gene a mutation is located, but cryo-EM can reveal the effect of this mutation on the axoneme. For example, in a scenario where a point mutation in *ODAD1* caused ODAs to be misoriented on the axoneme, only cryo-EM/ET would be capable of revealing the disease mechanism as the TEM would appear normal. Hence, genetics and 3D reconstructions are both important for describing detailed disease mechanisms. To make the point

about the importance of genetics, we have added the following sentence to the final discussion section “We have shown that the translational potential of this approach is particularly powerful when molecular genetics are combined with cryo-EM”. It is for this reason that our proof-of-concept study deliberately used mutations that were genetically defined. Testing whether structural defects in cryo-EM reconstructions can directly identify mutated genes or variants was not the intention of this study, and as the reviewer suggests, would have required a completely different study design.

2) The text jumps between Chlamy and human data. Is Extended Data Fig.1 entirely Chlamy- if so state?
b. jumps to human- it is very confusing. Extended Fig. 3 same- likely Chlamy as a-HC is not shown.

We have attempted to divide the paper as much possible into 3 sections: the first describing the *Chlamydomonas* axoneme, the second describing the human axoneme, and the third reporting the effect of PCD-causing mutations on axoneme structure. We realize that the Extended Data figures didn't show the same organization. To better match the text, we have reorganized the Extended Data figures and broken Extended Data Fig. 1 into two different figures (now Extended Data Fig. 1 and 6). We have also checked and edited each figure legend to ensure that the organism is explicitly stated. We hope this will help to clarify the distinction between *Chlamydomonas* and human data presented in the paper.

3) Why not show that the 24 nm spacing of ODAs is unaffected in the CCDC39/40 mutants axoneme averages- you say in the text that these are ‘unaffected’ why not show it graphically along a longitudinal view- is the periodicity even subtly altered without the 96nm ruler (Figure 7c)? This work is something that could actually resolve a lot of this. Are both of these ‘rulers’ in fact completely independent?

We have added longitudinal panels to Fig. 7 showing that the 24-nm periodicity of the ODA-DC in both the *CCDC39* and *CCDC40* mutant structures is unaffected by the loss of the 96-nm periodicity and remain in frame with the internal 48-nm repeat. All evidence suggests that these “rulers” establish themselves independently.

4) You make a grand statement in the discussion about ‘help de-orphanize dynein HCs in other species’ but what about within the same species? Can you detect the ‘compartmentalization’ proximal-distally of your ODA heavy chains in the human respiratory cilia- the change from DNAH11 to DNAH9? Supplementary Data Table 4 suggests you have only DNAH9- discuss? Do other such regional differences in MIPS or RSs etc?

Our structural information can indeed be used to classify dynein HCs both within and between species. For example, helping to distinguish whether a dynein heavy chain is more likely to form IDAd or IDAc based on differences in their N-terminal tail domains.

However, distinguishing between DNAH11 and DNAH9 is challenging due to their distal-proximal compartmentalization and structural similarity. Although both proteins are present in our sample, we cannot separate them based on our current classification strategies. We have modeled DNAH9 as it's slightly more abundant than DNAH11 in our mass spectrometry analysis and it localizes to the distal sections of respiratory ciliary axonemes (Fliegauf *et al.*, 2005; 10.1164/rccm.200411-1583OC), which are most prevalent in our splayed axoneme sample.

To distinguish between DNAH11 and DNAH9, higher resolution maps or an approach like cryo-ET where the distal and proximal regions of the axoneme can be processed separately would be needed. This

approach will also be needed to discover if there are other regional differences, or differences among the nine DMTs. Our atomic models will help interpret these future studies.

Minor points

1) For clarity, when first describing the IDAs (end of page 2) maybe explicitly state that there are 6 single headed dyneins and a single double headed one which each appear within the 96 nm repeat once. For non-specialists, this is dense and tricky going.

We thank the reviewer for this suggestion and have edited the introduction to state “Within a 96-nm repeat, there are six different single-headed inner dynein arms (IDAa–e and IDAg), one double-headed inner dynein arm (IDAf), and four identical multi-headed outer dynein arms (ODAs)”.

2) For a general audience, it may be worthwhile stating what the levels of resolution are within the field prior to this study (end of page 3). What resolution is necessary for the ‘most complete atomic model of the axoneme to date’?

Earlier cryo-ET studies of the *C. reinhardtii* axoneme have achieved at best 25 Å resolution, which is likely an overestimation due to the 0.5 FSC criterion used for subtomogram averages. As shown in the table below, all our maps achieve higher resolution than these studies using the 0.143 FSC criterion. We have added a statement to that effect to the manuscript.

Complex	Our resolution	Best cryo-ET	EMDB ID	Reference
N-DRC	3.2 - 8.2 Å	27 Å	20338	Gui et al. , 2019; 10.1073/pnas.1910960116.
T/TH	3.0 - 16.7 Å	25 Å	12162	Kutomi et al. , 2021; 10.1126/sciadv.abf3621
IDAf	8.4 - 10.2 Å	38 Å	20566	Fu et al. , 2021; 10.1096/fj.202001857R

Similarly, the best previous resolution for a human axoneme from respiratory cilia is 33.6 Å (Zhao *et al.*, 2021; 10.1091/mbc.E20-12-0806). As shown in Extended Data Table S7a our resolutions range from 3.3 Å (for the doublet microtubule) to 13.4 Å (for the head of radial spoke 3).

It is not only our improved resolution maps that have allowed us to generate the most complete atomic model of the axoneme to date, but revolutionary improvements in structure prediction (e.g., AlphaFold2) and the large body of prior work this study was built on. In Supplemental Table 2 we attempt to describe all the information that has been used to build our atomic models.

3) P6, ‘a 155-Å tall tower’ is too colloquial and confusing for this dataset.

We have edited the text to describe it as a “a 155-Å tall armadillo repeat protein” to be more precise.

4) P8, human respiratory cilia there are 6 isoforms of human actin. Does your modelling tell you which one it is for IDAa and IDAc?

In our proteomic analysis of human axonemes (Gui *et al.*, 2022; 10.1073/pnas.2207605119), we identified three of the eight isotypes: ACTA2, ACTG1 and ACTC1. Of these, ACTA2 is the most abundant and the one we modeled in the IDA structures. However, our map resolution is insufficient to rule out either ACTG1 and ACTC1, which are 94 and 98% identical to ACTA2. It is possible that different IDAs may

contain different actin isoforms or that there is redundancy among the actins. Addressing this issue would require further work analyzing cells with actin isoform deficiencies. We have added a note to Supplementary Table 4 explaining why ACTA2 was modeled.

5) P9, when you speak of coding mutations like the 'c.853G>A substitution in *ODAD1*' I believe the *ODAD1* needs to be italicized, likely for all the genes in this paragraph.

Thank you for the comment, we have amended the text so that we use capital italics when referring to the gene mutations.

6) P9, the text reads awkwardly 'by patient genome sequencing analysis' by 'analysis of patient genome sequencing'.

We have amended the text accordingly.

7) Docking complex mutants should not affect cilia number- why do the authors state 'too few cilia could be obtained for imaging'. In contrast, CC3C39/40 mutants have as their table says lots of microtubule disorganization and one would have thought would be more difficult to image- this is not the case, they are higher resolution. Can the authors explain why?

We were unable to obtain as many cilia for the *ODAD1* mutants as the ALI cultures did not grow as well, leading to a lower cilia harvest and therefore fewer DMTs on the cryo-EM grid. Although we did not quantify the number of cilia per cell, we do not think that *ODAD1* mutants had fewer cilia based on visual inspection.

8) In your discussion, you highlight 'striking species-specific differences'- you undersell the fact your methodology could be applied to look at cilia-specific differences within the same species, say airway cells and ependymal cilia lining the brain, highlighting the possible molecular foundations for differences to sensitivity to dysfunction when specific genes are mutated.

We have added a sentence to this effect at the end of the discussion. We think cryo-EM could be a powerful approach to reveal why some cilia types are more sensitive to mutations than others.

9) Figure 7a- how many samples, what is plotted? Is this an average? Statistics on the humans should be referenced within figure legends.

These values are averages calculated from videos. For the CCDC40, CCDC39, *ODAD1* (s) and *ODAD1* (n) mutants we used 8, 6, 3, and 8 videos respectively. These sample numbers have been added to the figure legend.

10) Extended Data Fig.2a- why are the numbers of particles lower (and lower resolution for IDAb) given the repeats should all be fairly similar in how they are averaged?

For each axonemal complex, we performed 3D classification to isolate particles with and without the complex. IDAb appears to dissociate readily from DMTs, as observed in an independent cryo-ET study of human respiratory cilia (Lin *et al.*, 2014; 10.1038/ncomms6727), explaining why its structure was reconstructed from fewer particles than other, more stably bound complexes.

11) Extended Data Fig 4. Love the London tube map but explain why some lines end without a 'station' complex- i.e. IDAb on 2 RS3S on 6?

The lines represent the paths taken by surface-bound coiled coils. Some axonemal complexes (like IDAb) dock onto the middle of the coiled coils instead of at their termini, explaining why some lines end without a station. We have edited the figure legend to make this clearer.

12) What does 'Epithelial disruption score' mean in Extended Data Fig.7c Y axis label?

The Epithelial disruption score is based on Thomas *et al.*, 2009; 10.1183/09031936.00153308. Scores between 1 and 4 were assigned to each video, where 1=normal cells, 2=minor disruption, 3=major disruption, and 4=single cells.

13) Extended Data Fig. 8: CCDC40 patient ID125 and ODAD1 ID25 patient sections are too dark to resolve any of the ultrastructure- need to show features being highlighted.

We have altered the contrast of these figures (now Extended Data Fig. 9), which has improved the visualization of the ultrastructure. We have also added labels to the control to highlight the normal appearance of the axonemal complexes and more arrows to highlight the defects in the mutant axonemes. To maintain patient anonymity, we have also had to change the IDs of the patient samples.

Edits

1) Typo Methods: Clinical transmission electron microscopy- should be 'Sorenson's'

We have corrected the typo.

2) Extended Data Fig.7 legend 'PCD CCDC40 mutation patient' awkward- consider 'PCD patient with CCDC40 mutation'?

Thank you. We have changed the awkward phrasing.

Referee #3 (Remarks to the Author):

The manuscript reports the first nearly complete structural model of the 96-nm repeat of the axoneme from *Chlamydomonas reinhardtii* flagella with all the major structural components included. The whole model includes the pseudo-atomic structures of the inner dynein arms (IDAs) and nexin-dynein regulatory complexes (N-DRCs) that are newly determined in the manuscript by single particle cryo-EM, in addition to previously determined structures of outer dynein arms (ODAs), the microtubule tubule doublet complex, the radial spoke complex and the central pair complex that were published by the authors' lab and others. The structural map calculations were nicely done – they utilized single particle cryo-EM with local masks to reconstruct the axonemal component complexes. Some of the structural model building referenced the protein structures predicted by AlphaFold. The new structural information provides important insight into the dynein docking and beating regulatory mechanisms that involve DRC and the newly identified outer-inner dynein links.

Moreover, the manuscript determined the pseudoatomic models of axonemal 96nm repeats for human

respiratory cilia from four patients of primary ciliary dyskinesia (PCD), which could offer direct structural interpretation on how the PCD mutations result in disease phenotypes. The structural comparison of the human and *Chlamydomonas* axonemes also revealed some species-dependent structural differences, particularly in IDAs. These comparisons provide new data and insights for understanding motile cilia evolution. Overall, the manuscript presents excellent structural biology work with a large amount of research effort involved. The whole structural models of the 96 nm repeats of motile cilia will serve as structure baselines for further mechanistic understanding of motile cilia beating and regulation. In addition, the paper itself is well-organized and clearly written. I recommend it for publication. I have no major concern except for the following minor suggestion to the authors.

This is a comprehensive structural report for a complicated molecular machine with many proteins and protein complexes. For the human axonemes, as summarized in Extended Data Fig 2a, different structural maps are shown with different resolutions ranging from 3.3 to 13.4Å. It would be helpful if another column is added behind the resolution column to give a brief description of the methods used for model building with each of the maps. Such a table should also be provided for the axoneme of *Chlamydomonas* flagellar structural maps and model building because it would help the readers develop an accuracy expectation for each part of the whole structural model.

We thank the reviewer for their suggestion. We have added a new table to Extended Data Fig. 1 (panel b) which describes each individual complex of the *Chlamydomonas* axoneme, the resolution it was determined to, and the method or methods used to model it. We have also generated an equivalent table for the human data (Extended Data Fig 7a).

For *Chlamydomonas*, further details of the modeling process are provided in Supplementary Table 2, which describes the modeling rationale for each subunit identified in this study, and explicitly states the use of *de novo* and AlphaFold2 modeling.

Reviewer Reports on the First Revision:

Referees' comments:

Referee #1 (Remarks to the Author):

Walton et al. addressed points from the reviewers or at least clarified their position convincingly. This reviewer has only a few further points. This manuscript should be published in Nature after they are fixed. This work has great impact on cilia and cellular motility research community, molecular and cellular biology of motor proteins.

Global dynein conformations

Thank you for clarifying the way to model ODA and MTBDs of dyneins. This part in Methods was much more improved. However, for non-experts, there can be still confusion about how the entire model was built. In the images like Fig. 2a, Fig. 4f, Fig.5b, the readers will have an impression that the whole model was built as one single particle cryo-EM reconstruction. In reality, each model is a hybrid. Are the stalk/MTBD models in these panels from past SPA works of isolated dyneins fitted to your SPA of ODA/IDA anchored on DMT or Alphafold2 prediction? How were the two structures (main body of dynein and MTBD) aligned and docked? This should be described clearly, maybe both in the text and figure captions. Many readers interested in motor proteins will see these images to know how the orientation of MTBDs in all the ODAs/IDAs – if they are oriented by hybrid modeling, the readers should be informed. Probably the same for RS (Fig.5c). If this reviewer understands correctly, the base part of RS was modelled by your present SPA work of sprayed axoneme, while the head part was merged as hybrid with your past SPA of isolated RS. Here also this reviewer recommends to add statements in the caption.

Supplementary Table 1

For those who are interested in cilia from other species and refer this phenomenal paper for their researches, it will be kind to show protein IDs, such as Uniprot id, accession or gi, which can be read by blast. Blast does not read Phytozome ID.

Referee #2 (Remarks to the Author):

There was very little to fault and much to applaud with the initial submission, and generally the revised manuscript by Walton, Gui et al has integrated suggestions to improve accessibility of the beautiful and important work to wider audiences. My only outstanding reservations are to the human data that should be addressed before publication in Nature:

(1) New extended data figure 9a- TEM images for ODAD1 ID02 (2/3 rightmost should be removed as out of focus) and CCDC40 (ID03) (almost all are out of focus, should be replaced). These phenotypes are described elsewhere but these are orientations most are used to looking at- and a picture should be chosen to best illustrate a phenotype (i.e. representative). Many of these are not useful. A clear, in focus panel should be chosen if really >100 screened cross sections were available. Minor point: watch where your missing ODA arrowheads are in ID01 and ID02 I think they are pointing to the wrong spot on several DMTs. In b, a comparative image showing control human doublet MT with 96 nm periodicity of RS maintained would be useful to show readers what the structure should look like, not just how it is disturbed on mutation.

(2) Supplementary table 6: I would strongly recommend hyperlinking to OMIM as a column to

future-proof work where no disease association yet exists but also helps track multiple previous names as well as published work on human genetics and animal models, for example TUBA1A OMIM:602529- hyperlinking would be perfect! It takes time to update (like a WIKI site) but it will happen- CFAP21 has many other names it has been known by OMIM: 617906 but the disease association is not there yet. Where an OMIM term does not exist- you can leave the cell blank and leave your references in. Others could just rely on OMIM? It is really helpful for landmarking to previous work.

Minor points:

- (1) New sentence penultimate paragraph p.3 'The resolution of these maps exceeds what has been achieved by cryo-ET of *C. reinhardtii* axonemes.'- maybe include a range like you put in the rebuttal and the reference there?
- (2) Figure 4 legend- state explicitly in the legend that the 8 nm size marker represents one tubulin heterodimer? It's in the text of the methods but would be helpful here.
- (3) Figure 6b legend- state that it is cropped below the heavy chain AAA domains- only a portion of the reconstruction on the DMTs? And that the densities outside this region in the dynein tail domain are sufficient to allow pinpointing of which DHC/DNAH they are?
- (4) Figure 7 legend- explain what the blue 48 nm ruler is in words.
- (5) Methods for ciliary beat analysis- please enter how epithelial disruption score is calculated in the text. It is not intuitive.
- (6) Extended data figure 5- bottom of your image panel is corrupted/cropped.
- (7) Extended data figure 6a- include age (can be range no help anonymize patients) and sex. You focus on MMAF as phenotypes but that will only affect half patients and only of a certain age.
- (8) Extended data figure 8: Numbers of movies used for quantification are provided- but from how many inserts (technical repeats) or just one per patient.
- (9) Respiratory epithelial culture and cilia isolation: missing details on total numbers of cells used for the experiment and how many isolations from different cultures (1 set of cultures per individual or multiple experiments)- to construct these datasets- this will be important for people.

Edits:

- (1) Page 4 first paragraph- 'closely resembles the core of ODA' – plural ODAs?
- (2) You generally write 'coiled coil' but 5 times in the documents write coiled-coil- be consistent? i.e. p.10 'MIA and the coiled-coil of the ODA-DC may', p11 'in the identity of the coiled-coil docking factors'

Author Rebuttals to First Revision:

Referee #1 (Remarks to the Author):

Walton et al. addressed points from the reviewers or at least clarified their position convincingly. This reviewer has only a few further points. This manuscript should be published in Nature after they are fixed. This work has great impact on cilia and cellular motility research community, molecular and cellular biology of motor proteins.

Global dynein conformations

Thank you for clarifying the way to model ODA and MTBDs of dyneins. This part in Methods was much more improved. However, for non-experts, there can be still confusion about how the entire model was built. In the images like Fig. 2a, Fig. 4f, Fig.5b, the readers will have an impression that the whole model was built as one single particle cryo-EM reconstruction. In reality, each model is a hybrid.

To avoid giving the impression that the entire model was built into a single cryo-EM map, we state in the main text that “Due to computational limits imposed by the large size of the repeat, we focused on individual axonemal complexes by particle subtraction, refinement, and classification before reconstituting the full 96-nm repeat *in silico*.” This is reiterated in the methods, and in the SI, where we have devoted over 10 figures to the processing of individual complexes in the 96-nm repeat.

Are the stalk/MTBD models in these panels from past SPA works of isolated dyneins fitted to your SPA of ODA/IDA anchored on DMT or AlphaFold2 prediction? How were the two structures (main body of dynein and MTBD) aligned and docked? This should be described clearly, maybe both in the text and figure captions. Many readers interested in motor proteins will see these images to know how the orientation of MTBDs in all the ODAs/IDAs – if they are oriented by hybrid modeling, the readers should be informed.

To better understand how this was modeled, we have included the following into the Methods section:

“Homology models of the dynein motor domains (including the linker, AAA+ modules, stalk, and MTBD) were generated in the post-powerstroke state using SWISS-MODEL⁷³, as they were too large to be modeled by AlphaFold2.”

“Homology models of motor domains in the post-powerstroke conformation were positioned by fitting into the observed cryo-EM densities for the linker, individual AAA+ modules, and when possible, the base of the stalk.”

Probably the same for RS (Fig.5c). If this reviewer understands correctly, the base part of RS was modelled by your present SPA work of sprayed axoneme, while the head part was merged as hybrid with your past SPA of isolated RS. Here also this reviewer recommends to add statements in the caption.

We find that this information is best fit in the main text and methods to allow a full description, so that no confusion can arise from an abbreviated treatment of this topic in the limited space of a caption. In the main text, methods, and SI, we currently make explicit reference to the models used to assemble to full 96-nm repeat.

Supplementary Table 1

For those who are interested in cilia from other species and refer this phenomenal paper for their researches, it will be kind to show protein IDs, such as Uniprot id, accession or gi, which can be read by blast. Blast does not read Phytozome ID.

We have now added Uniprot or NCBI accession numbers for the sequence that are identical to the Phytozome entries.

Referee #2 (Remarks to the Author):

There was very little to fault and much to applaud with the initial submission, and generally the revised manuscript by Walton, Gui et al has integrated suggestions to improve accessibility of the beautiful and important work to wider audiences. My only outstanding reservations are to the human data that should be addressed before publication in Nature:

(1) New extended data figure 9a- TEM images for ODAD1 ID02 (2/3 rightmost should be removed as out of focus) and CCDC40 (ID03) (almost all are out of focus, should be replaced). These phenotypes are described elsewhere but these are orientations most are used to looking at- and a picture should be chosen to best illustrate a phenotype (i.e. representative). Many of these are not useful. A clear, in focus panel should be chosen if really >100 screened cross sections were available. Minor point: watch where your missing ODA arrowheads are in ID01 and ID02 I think they are pointing to the wrong spot on several DMTs.

We have removed the out-of-focus TEM images and improved the labeling. We now have a single, representative example for each mutant and a non-PCD control.

In b, a comparative image showing control human doublet MT with 96 nm periodicity of RS maintained would be useful to show readers what the structure should look like, not just how it is disturbed on mutation.

We thank the reviewer for the suggestion. We have now added a comparative image showing the regular spacing of radial spokes in a non-PCD control.

(2) Supplementary table 6: I would strongly recommend hyperlinking to OMIM as a column to future-proof work where no disease association yet exists but also helps track multiple previous names as well as published work on human genetics and animal models, for example TUBA1A OMIM:602529- hyperlinking would be perfect! It takes time to update (like a WIKI site) but it will happen- CFAP21 has many other names it has been known by OMIM: 617906 but the disease association is not there yet. Where an OMIM term does not exist- you can leave the cell blank and leave your references in. Others could just rely on OMIM? It is really helpful for landmarking to previous work.

We thank the reviewer for the suggestion and have added OMIM values to Supplementary Table 5. We have kept the original references rather than just relying on OMIM values.

Minor points:

(1) New sentence penultimate paragraph p.3 'The resolution of these maps exceeds what has been achieved by cryo-ET of *C. reinhardtii* axonemes.'- maybe include a range like you put in the rebuttal and the reference there?

We now mention the highest resolution cryo-ET reconstruction of the *C. reinhardtii* axoneme (25 Å) and have added its reference.

(2) Figure 4 legend- state explicitly in the legend that the 8 nm size marker represents one tubulin heterodimer? It's in the text of the methods but would be helpful here.

We have added an explanation of the scale bar.

(3) Figure 6b legend- state that it is cropped below the heavy chain AAA domains- only a portion of the reconstruction on the DMTs? And that the densities outside this region in the dynein tail domain are sufficient to allow pinpointing of which DHC/DNAH they are?

We now describe in the legend that the image is cropped and that we are only showing the bases of the IDA HCs.

(4) Figure 7 legend- explain what the blue 48 nm ruler is in words.

We have expanded the legend to explain that the blue densities correspond to MIPs.

(5) Methods for ciliary beat analysis- please enter how epithelial disruption score is calculated in the text. It is not intuitive.

We have added a methods section that explains how epithelial disruption score was calculated and cites the original paper.

(6) Extended data figure 5- bottom of your image panel is corrupted/cropped.

To avoid errors in PDF conversion, we have now supplied our Extended Data Figures as JPEGs.

(7) Extended data figure 6a- include age (can be range no help anonymize patients) and sex. You focus on MMAF as phenotypes but that will only affect half patients and only of a certain age.

We have added age and sex information to the table (now Extended Data Table 2).

(8) Extended data figure 8: Numbers of movies used for quantification are provided- but from how many inserts (technical repeats) or just one per patient.

For each mutation, the movies (of respiratory cilia from nasal brushings) come from the same patient. We have added this information to the figure legend.

(9) Respiratory epithelial culture and cilia isolation: missing details on total numbers of cells used for the experiment and how many isolations from different cultures (1 set of cultures per individual or multiple experiments)- to construct these datasets- this will be important for people.

We have added the total number of inserts used (12), the size of the insert, and the number of cells seeded onto them. Deciliation was repeated 6 times to maximize the yield of cilia before combining. Each cryo-EM reconstruction of a mutant axoneme was generated from a single plate (12 inserts).

Edits:

(1) Page 4 first paragraph- 'closely resembles the core of ODA' – plural ODAs?

We have modified this sentence.

(2) You generally write 'coiled coil' but 5 times in the documents write coiled-coil- be consistent? i.e. p.10 'MIA and the coiled-coil of the ODA-DC may', p11 'in the identity of the coiled-coil docking factors'

We have modified the text to only use coiled coil.